# Cell-inspired design of cascade catalysis system by 3D spatially separated active sites

Qiuping Wang[1,2,6], Kui Chen[3,6], Hui Jiang[4,6], Cai Chen[1], Can Xiong[1], Min Chen[1], Jie Xu[5], Xiaoping Gao ®[2] ✉, Suowen Xu ®[1] ✉, Huang Zhou ®[1] ✉ & Yuen Wu ®[1,2] ✉

Cells possess isolated compartments that spatially confine different enzymes, enabling high-efficiency enzymatic cascade reactions. Herein, we report a cell-inspired design of biomimetic cascade catalysis system by immobilizing Fe single atoms and Au nanoparticles on the inner and outer layers of three-dimensional nanocapsules, respectively. The different metal sites catalyze independently and work synergistically to enable engineered and cascade glucose detection. The biomimetic catalysis system demonstrates ~ 9.8- and 2-fold cascade activity enhancement than conventional mixing and coplanar construction systems, respectively. Furthermore, the biomimetic catalysis system is successfully demonstrated for the colorimetric glucose detection with high catalytic activity and selectivity. Also, the proposed gel-based sensor is integrated with smartphone to enable real-time and visual determination of glucose. More importantly, the gel-based sensor exhibits a high correlation with a commercial glucometer in real samples detection. These findings provide a strategy to design an efficient biomimetic catalysis system for applications in bioassays and nanobiomedicines.

Live cells contain multiple compartmentalized organelles surrounded by membranes, which enable the spatial confinement of enzymes in different cellular domains, shielding them from each other and allowing enzymatic cascade reactions to occur in optimized microenvironments with high activity and specificity[1,2]. This inspires researchers to simulate multicompartmental living cellular systems for distinct biochemical reactions in one pot by encapsulating different enzymes in specific positions[3,4]. However, protein enzymes not only exhibit unsatisfactory stability owing to their inherently fragile nature but also suffer from high costs and difficulties in recovery and storage, which significantly impedes their practical industrialization.

Natural metalloenzymes contain specific metal ions (such as Fe, Ni, and Cu) that act as active sites and can catalyze a wide range of many important biological and chemical reactions[5,6]. Recently, nanomaterials (such as metal nanoparticles, nanoclusters, and single atoms) with active metal sites have exhibited excellent stability and controllable activity in many challenging biochemical transformations[7–9], which are recognized as potential substitutes for natural metalloenzymes. However, these metal sites usually exhibit only one type of enzyme-mimicking activity in a catalytic system. This indicates that the integration of different active sites together is essential to complete multistep reactions in a cascade catalytic system. For instance, hybrid

[1]Department of Endocrinology, The First Affiliated Hospital of USTC, Division of Life Sciences and Medicine, University of Science and Technology of China, Hefei 230001, China. [2]Hefei National Laboratory for Physical Sciences at the Microscale, School of Chemistry and Materials Science, University of Science and Technology of China, Hefei 230026, China. [3]Key Laboratory of Strongly Coupled Quantum Matter Physics, Chinese Academy of Sciences, School of Physical Sciences, University of Science and Technology of China, Hefei 230026, China. [4]Department of Cardiothoracic Surgery, The First Affiliated Hospital of USTC, Division of Life Sciences and Medicine, University of Science and Technology of China, Hefei 230001, China. [5]College of Chemistry and Materials Engineering, Wenzhou University, Wenzhou 325035, China. [6]These authors contributed equally: Qiuping Wang, Kui Chen, Hui Jiang.
✉e-mail: gaoxiaoping2014@foxmail.com; sxu1984@ustc.edu.cn; huangz02@ustc.edu.cn; yuenwu@ustc.edu.cn

cascade catalysts that combine glucose oxidase (GOx)-like and horseradish peroxidase (HRP)-like active sites have been developed and utilized in glucose detection[10,11]. The traditional strategy involves a simple stacking of different sites by coplanar construction, such as loading Au NPs with GOx-like activity on the surface of peroxidase-mimicking nanomaterials for cascade glucose detection[10,12,13], but these systems cannot effectively isolate the catalytic sites and prevent interference from each other, resulting in a low cascade reaction efficiency. Therefore, a three-dimensional (3D) highly spatially separated distribution of different sites and corresponding noninterfering reaction pathways (as in living cellular systems) are expected to design an efficient cascade catalysis system.

Herein, we present a cell-inspired design of a biomimetic cascade catalysis system by integrating Fe single atoms (SAs) and Au nanoparticles (NPs) into different layers of N-doped carbon (NC)-based 3D nanocapsules, which enables cascade catalytic reactions with noninterference and high efficiency. The monodispersed Fe SAs fixed on the interior surface of the nanocapsules through thermal diffusion show peroxidase (POD)-like activity. Au NPs loaded on the outer surface show intrinsic GOx-like activity. The obtained biomimetic catalysis system demonstrates cascade activity enhancement compared to its counterparts prepared by conventional mixing and coplanar construction. Theoretical calculations further unveil that the suitable adsorption of intermediates can lead to a low reaction energy barrier to enhance the POD-like performance in the cascade reaction. More

importantly, the biomimetic catalysis system shows high activity and stability in glucose cascade catalytic colorimetric sensors. Furthermore, the proposed gel-based sensor was combined with a smartphone to realize the real-time and visual determination of glucose.

## Results and discussion

Inspired by eukaryotic cells, we developed a cell-stimulated design of a biomimetic cascade catalysis system as illustrated in Fig. 1a. In this system, Fe SAs and Au NPs are separately confined and closely positioned at distinct layers of 3D nanocapsules to realize efficient cascade catalysis. Peanut-shaped $\alpha$-$Fe_2O_3$ with an average diameter of 524 nm was firstly synthesized by a hydrothermal method (Fig. 1b and Supplementary Fig. 1). A polydopamine (PDA) layer was subsequently deposited onto the surface to form a core-shell architecture, denoted as $Fe_2O_3$@PDA (Fig. 1c and Supplementary Fig. 2). Then, the $Fe_2O_3$@PDA was annealed under an argon (Ar) atmosphere, during which the PDA layer was transformed in situ to form N-doped carbon shells after carbonization. The depressed signal for the organic species (N/O−H) in the Fourier transformed infrared (FT-IR) spectroscopy and the appearance of characteristic D (1346 cm$^{-1}$) and G (1588 cm$^{-1}$) bands of carbon species in Raman spectra both traced this evolution (Supplementary Fig. 3)[14]. Meanwhile, the $Fe_2O_3$ on the surface was gradually reduced to metal Fe by carbon, accompanied by the Fe atoms diffusing locally in the heat drive and being trapped by N defects on NC shells due to the strong interaction (Supplementary Fig. 4). Acid leaching was

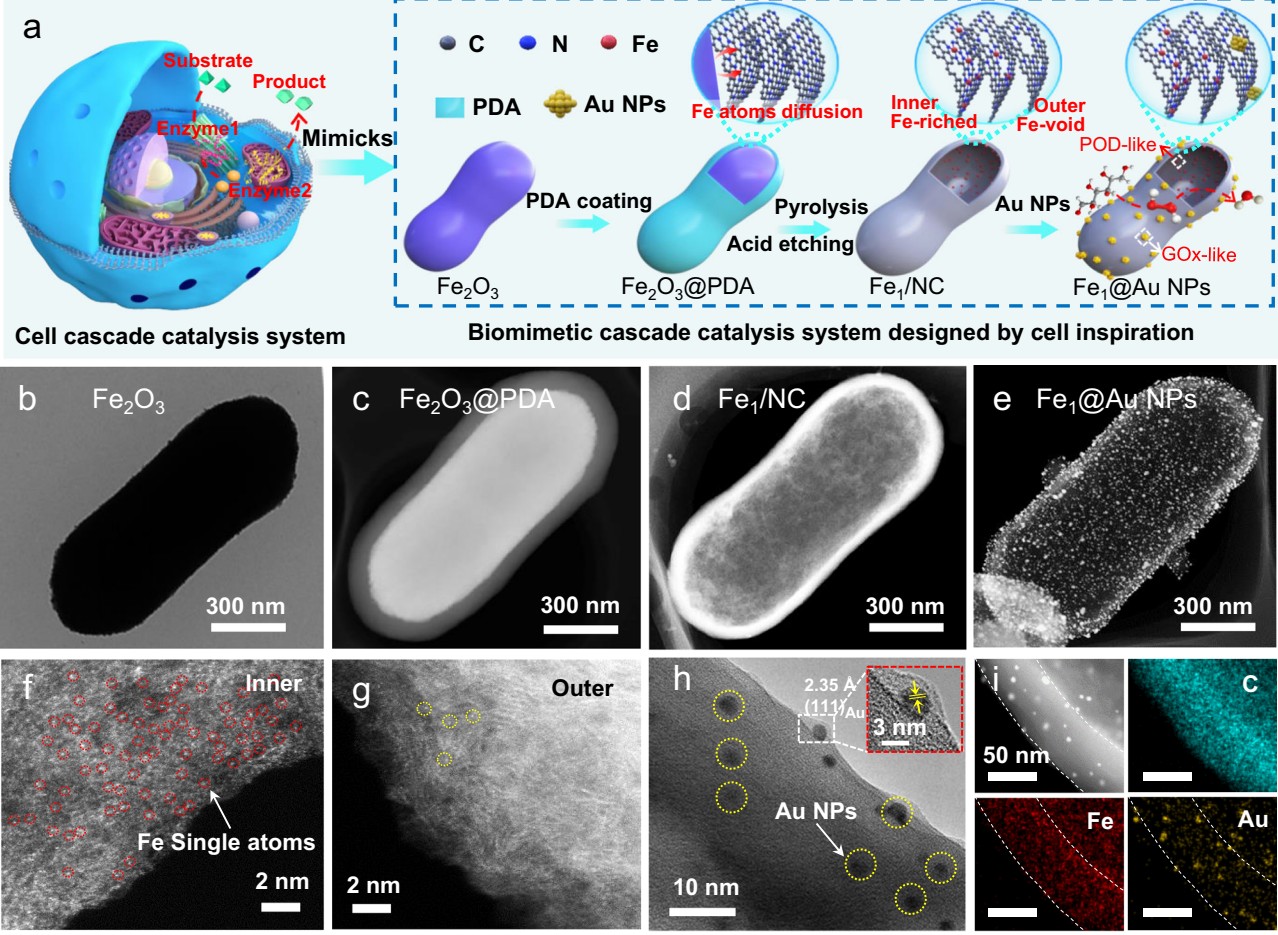

**Fig. 1 | Schematic illustrations and morphology characterization for the preparation of Fe₁@Au NPs and reference materials. a** Schematic illustration of the cell cascade catalysis system and biomimetic cascade catalysis system. **b** TEM image of Fe₂O₃. **c**–**e** HAADF images of Fe₁@Au NPs and the reference materials. **f**, **g** Atomic-resolution HAADF-STEM images of Fe₁/NC. **h** HRTEM image of Fe₁@Au NPs. Inset: magnified HRTEM image of Au NPs. **i** EDS mappings of Fe₁@Au NPs. All experiments were independently repeated three times with similar results.

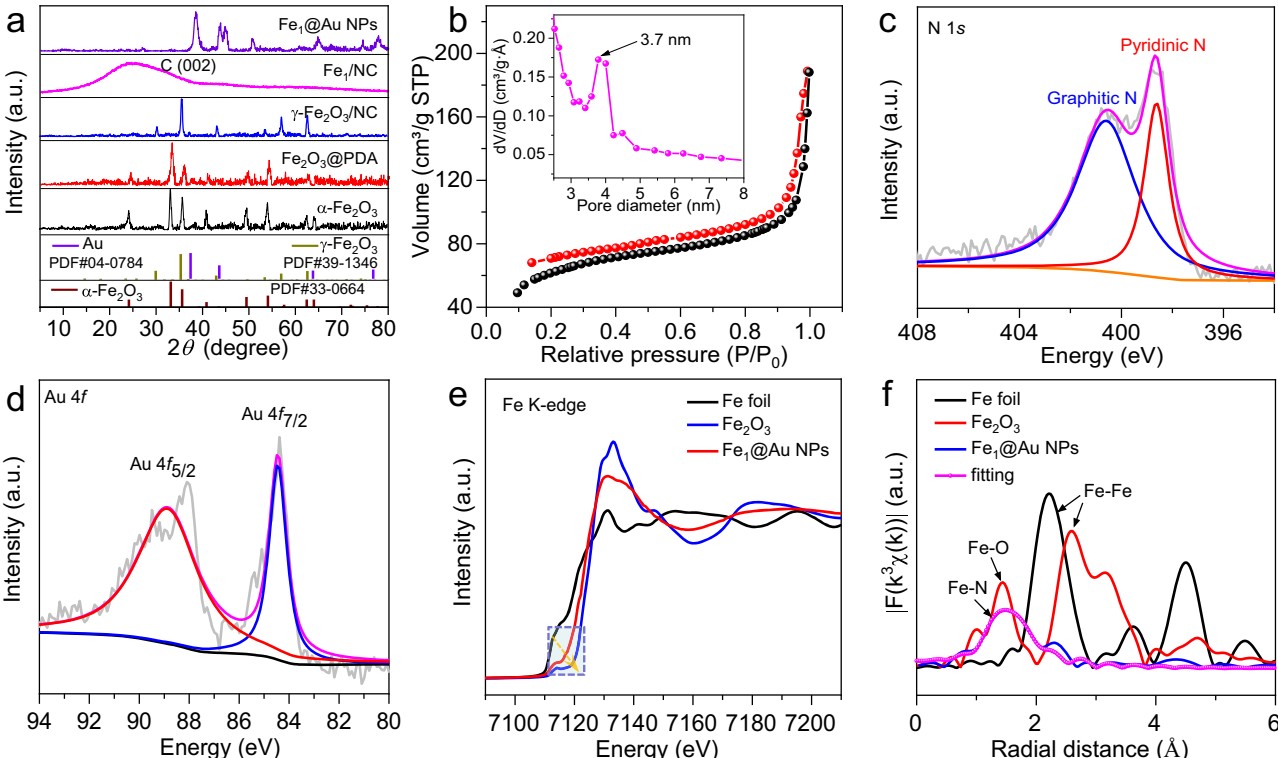

**Fig. 2 | Structural characterizations of Fe₁@Au NPs and reference materials.** **a** XRD patterns. **b** N₂ adsorption isotherm and the corresponding pore-size distribution of Fe₁@Au NPs. **c** N 1s and **d** Au 4f XPS spectra of Fe₁@Au NPs. **e** Fe K-edge NEXAFS spectra. **f** Fourier transformed (FT) k3-weightedχ(k)-function of the EXAFS spectra.

further introduced to remove the remaining $Fe_2O_3$ without thermal diffusion on $Fe_2O_3$@PDA, and stable Fe single atoms anchored on the inner surface of the NC shell were obtained (denoted as Fe₁/NC). Figure 1d and Supplementary Fig. 5 reveal that the as-obtained Fe₁/NC displays a 3D hollow nanocapsule structure with a microscale length and a diameter of 620 nm, and no Fe particles were observed. Finally, Au nanoparticles with an average diameter of 3.6 nm were uniformly loaded onto the outer surface of the synthesized nanocapsule to obtain a biomimetic catalysis system (denoted as Fe₁@Au NPs, Fig. 1e, h and Supplementary Fig. 6). Aberration-corrected high angle annular dark-field scanning transmission electron microscopy (AC HAADF-STEM) confirmed that high-density Fe atoms were mainly monodispersed on the inner surface of the nanocapsules, but few were dispersed on the outer layer (Fig. 1f, g and Supplementary Fig. 7). Figure 1h further shows that Au NPs on the outer surface possess a typical lattice fringe of 0.23 nm attributed to the Au (111) plane. Energy dispersive X-ray spectroscopy (EDS) mappings confirm that Fe and Au species were homogeneously dispersed on the inner layer and outer layer of the nanocapsules (Fig. 1i), respectively. These results indicate that a unique catalysis system with a highly separated 3D spatial distribution of different metal sites was successfully constructed.

X-ray diffraction (XRD) patterns were used to trace the phase changes during the construction of the whole system. Figure 2a shows the characteristic peaks of $\alpha$-$Fe_2O_3$ in the initial stage. After the PDA coating and subsequent carbonization treatment, the depressed signal for $\alpha$-$Fe_2O_3$ and the appearance of $\gamma$-$Fe_2O_3$ indicate that the phase transition of $Fe_2O_3$ occurred at high temperatures. After acid etching, the typical signal of $\gamma$-$Fe_2O_3$ vanished, and the obtained Fe₁/NC shows a wide peak at $2\theta = 25.3°$ corresponding to the carbon (002) peak. Additionally, no characteristic Fe peaks can be detected, excluding the existence of large Fe metal and oxide aggregation in Fe₁/NC. Subsequently, after the deposition of Au NPs, the obtained Fe₁@Au NPs shows the typical signal of Au NPs, in which the signals at 38.2°,

44.4°, 64.5°, and 77.5° correspond to the (111), (200), (220), and (311) lattice planes, respectively (JCPDS No. 04-0784). The N₂ adsorption–desorption isotherm shows that Fe₁@Au NPs possesses a specific surface area up to 220.84 m² g⁻¹ and abundant mesopores (Fig. 2b), which can facilitate substrate and product molecule diffusion and electronic transport in catalytic reactions. The inductively coupled plasma–mass spectrometry (ICP-MS) results reveal that the Fe and Au contents of Fe₁@Au NPs are 6.39 wt% and 2.08 wt%, respectively (Supplementary Table 1). X-ray photoelectron spectroscopy (XPS) tests reveal that the C, N, Fe and Au contents are 86.4 at%, 4.4 at%, 0.4 at% and 0.8 at% on the Fe₁@Au NPs surface (Supplementary Table 1), respectively. The high-resolution XPS and near edge X-ray absorption fine structure (NEXAFS) spectra of N 1s reveal that two prominent peaks for Fe₁@Au NPs are assigned to pyridinic N (398.5 eV) and graphitic N (400.7 eV) (Fig. 2c and Supplementary Fig. 8). The pyridinic N can contribute one p-electron to the π conjugated system, favoring the stabilization of Fe single atoms[15].

XPS and XAFS were used to further characterize the valence state and detailed structural information of Fe₁@Au NPs. The binding energies of the Au $4f_{7/2}$ and Au $4f_{5/2}$ peaks appeared at 84.5 eV and 88.8 eV, respectively, suggesting that Au NPs are only composed of metallic $Au^0$ (Fig. 2d). The binding energies of the Fe $2p_{3/2}$ and $2p_{1/2}$ peaks were centered at 712.3 eV and 724.9 eV (close to $Fe^{3+}$), respectively, indicating positively charged Fe single atoms (Supplementary Fig. 9). This is further confirmed by the X-ray absorption near-edge structure (XANES) result (Fig. 2e). The position of the white-line peak for Fe SAs is located between the Fe foil and $Fe_2O_3$, indicating the presence of ionic $Fe^{\delta+}$ ($0 < \delta < 3$) in Fe₁@Au NPs. The extended X-ray absorption fine structure (EXAFS) of the R space of Fe SAs displays only one dominant peak at approximately 1.46 Å, which is attributed to the Fe-N bond[15–17], verifying the atomic dispersion of Fe in the Fe₁@Au NPs. Further fitting results confirm that the first coordination number of the central atom Fe is about 4, suggesting that the proposed local

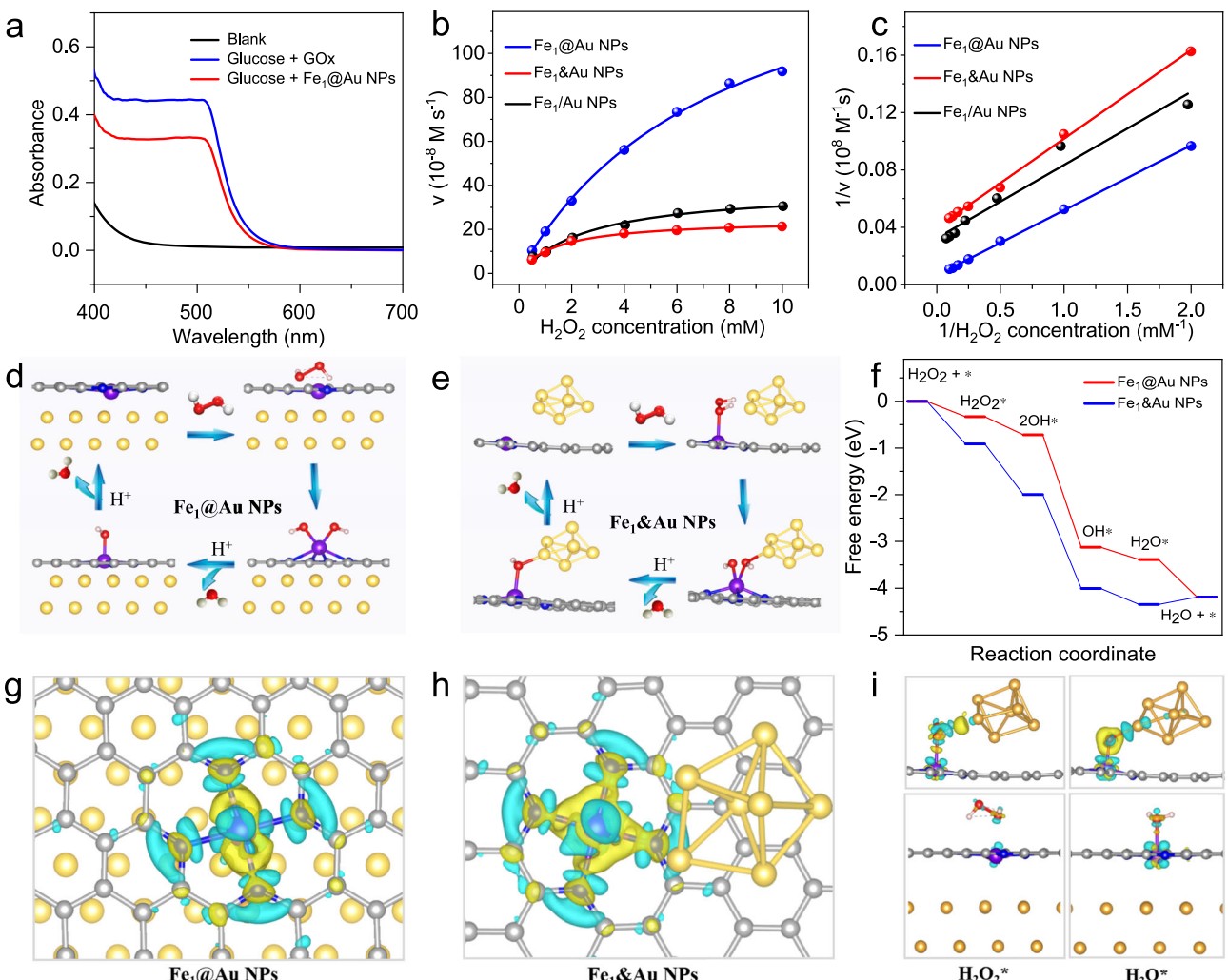

**Fig. 3 | The theoretical investigation of the mechanism for the enzymes-like reaction of Fe₁@Au NPs. a** The UV-Vis absorption spectra of the solutions obtained by gluconic acid-specific colorimetric assay. **b** Steady-state kinetic assay of peroxidase-like activity. Michaelis-Menten curves by varying $H_2O_2$ concentration at constant TMB concentration. **c** The corresponding Lineweaver-Burk plots with $H_2O_2$ as a substrate. The catalytic mechanism along the POD-mimicking reaction path on **d** Fe₁@Au NPs and **e** Fe₁&Au NPs. **f** The free-energy diagram for the POD-mimicking reaction. Charge density difference of **g** Fe₁@Au NPs and **h** Fe₁&Au NPs. **i** Charge density distributions of $H_2O_2^*$ and $H_2O^*$ groups absorbed on Fe₁&Au NPs (up) and Fe₁@Au NPs (down), where the isosurface value is set to be 0.005 e/Å³ and the positive and negative charges are shown in yellow and cyan, respectively. The gray, blue, purple, light yellow, red, and white balls represent C, N, Fe, Au, O, and H atoms, respectively.

structure is Fe-N₄. More fitting curves and parameters are shown in Supplementary Fig. 10 and Supplementary Table 2.

As the catalysis system (Fe₁@Au NPs) includes different catalytic metal species, it is most likely to catalyze two different reactions simultaneously to realize cascade catalysis. To verify this, two independent catalytic reactions were initially investigated, including GOx-mimicking and POD-mimicking catalysis reactions. Firstly, the Fe₁@Au NPs system can catalyze the oxidation of glucose to generate gluconic acid and $H_2O_2$ in the presence of $O_2$. The generated gluconic acid and $H_2O_2$ were verified by a specific colorimetric assay (Fig. 3a and Supplementary Fig. 11, more detailed analyses can be seen in Supplementary Information)[10,18–21]. This finding suggests the existence of GOx-mimicking activity within Fe₁@Au NPs. Additionally, Fe₁@Au NPs can catalyze the oxidation of 3,3',5,5'-tetramethylbenzidine (TMB), o-phenylenediamine (OPD) and 2, 2'-azino-bis(3-ethylbenzothiazoline-6-sulfonic acid) (ABTS) in the presence of $H_2O_2$, indicating that the biomimetic Fe₁@Au NPs system possesses the POD-mimicking activity[22] (Supplementary Figs. 12 and 13). The electron paramagnetic resonance (EPR) spectra also demonstrate the generation of glucose acid in GOx-mimicking and hydroxyl radical (·OH) intermediates in

POD-mimicking reactions (Supplementary Fig. 14), indicating that the Fe₁@Au NPs system can efficiently catalyze two reactions[23–25].

The POD-mimicking activity of Fe₁@Au NPs was systematically investigated by varying one substrate concentration while keeping the other constant. For a better comparison, one can use a convention mixing system for the preparation by physically mixing Fe SAs with Au NPs sites (denoted as Fe₁/Au NPs), or can prepare a coplanar system according to the previous reports (denoted as Fe₁&Au NPs, both Fe SAs and Au NPs sites are on the surface of the support, Supplementary Fig. 16)[26,27]. Fig. 3b, c and Supplementary Fig. 17 show that the steady-state kinetics followed the typical Michaelis–Menten model well in the tested concentration range of $H_2O_2$ and TMB. According to the fitted Lineweaver–Burk plots, the Km value of Fe₁@Au NPs was 1.83 for substrate $H_2O_2$, indicating the high affinity towards $H_2O_2$. Additionally, the Vmax values of Fe₁@Au NPs were calculated to be 166.33 and $92.37 \times 10^{-8}$ M s⁻¹ for $H_2O_2$ and TMB, respectively, which demonstrates that the biomimetic Fe₁@Au NPs system has a high peroxidase-like activity, exceeding that of Fe₁/Au NPs and Fe₁&Au NPs and most reported POD-mimicking catalysts-based systems (Supplementary Table 3).

Furthermore, density functional theory (DFT) computations were performed to reveal the origin of the outstanding POD-like activity of the biomimetic Fe$_1$@Au NPs. The POD-mimicking reaction pathways and the corresponding free energy diagram were analyzed (Fig. 3d–f and Supplementary Figs. 18 and 19)[28]. For both the biomimetic Fe$_1$@Au NPs and coplanar Fe$_1$&Au NPs models, H$_2$O$_2$ is easily end-on adsorbed on Fe sites with adsorption energies of −0.33 eV and −0.91 eV, respectively. Next, H$_2$O$_2$* dissociates to 2OH*, which is further reduced to H$_2$O*. Interestingly, all steps are exothermic processes for biomimetic Fe$_1$@Au NPs, in which the rate-determining step (RDS) of the POD-mimicking reaction is OH* protonation to H$_2$O* with an energy barrier of −0.26 eV. However, for coplanar Fe$_1$&Au NPs, H$_2$O* desorption in the final step is the RDS, requiring a large energy input of +0.16 eV due to the strong binding strength of the H$_2$O* species on the coplanar Fe$_1$&Au NPs. Furthermore, the charge density difference was determined to further explain the strong interaction between the H$_2$O* intermediates and coplanar Fe$_1$&Au NPs. Figure 3g, h show the charge accumulation and depletion mainly occurring on the Fe-N bonds and Fe or C atoms for the Fe$_1$@Au NPs and Fe$_1$&Au NPs models. A further study of intermediate adsorption (H$_2$O$_2$*, 2OH*, OH*, and H$_2$O*) on Fe sites (Supplementary Figs. 20 and 21) shows obvious charge accumulation and depletion in O*-Fe species for the biomimetic Fe$_1$@Au NPs. While there is extra charge accumulation and depletion in O*-Au species for coplanar Fe$_1$&Au NPs, indicating the generation of O-Au bonds and leading to the strong binding strength of H$_2$O* (Fig. 3i)[29]. The above results indicate the suitable adsorption strength of intermediates for the biomimetic Fe$_1$@Au NPs, which would facilitate OH* protonation to H$_2$O* (RDS step) and consequently enhance POD-like activity in the catalytic reaction.

Based on the tandem enzyme activities of the biomimetic Fe$_1$@Au NPs system, glucose cascade catalysis was further explored (Fig. 4a and Supplementary Fig. 22). Supplementary Fig. 23 shows that the specific activity (SA) of the Fe$_1$@Au NPs system is 52.29 U mg$^{-1}$ in glucose cascade catalytic reactions. Figure 4b shows that the biomimetic Fe$_1$@Au NPs system has the highest cascade catalytic activity, which is ~ 9.8- and 2.0-fold higher than that of the conventional mixed Fe$_1$/Au NPs and coplanar Fe$_1$&Au NPs systems, respectively. Furthermore, a steady-state kinetics assay was conducted to evaluate the enzyme-like catalytic performance. The initial reaction rate of the biomimetic Fe$_1$@Au NPs system versus the glucose concentration follows the typical Michaelis–Menten behavior. According to the fitted Lineweaver–Burk plots, the calculated Vmax value of Fe$_1$@Au NPs is $113.5 \times 10^{-8}$ M s$^{-1}$ for glucose, showing that the biomimetic Fe$_1$@Au NPs system has a high catalytic activity for glucose (Supplementary Figs. 24 and 25 and Supplementary Table 4). This markedly enhanced activity can be attributed to unique spatial segregation sites of the biomimetic Fe$_1$@Au NPs system, in which Au NPs and Fe SAs sites are separately confined and closely positioned at distinct layers of 3D nanocapsules, which can effectively facilitate H$_2$O$_2$ transfer in tandem reactions (also known as the proximity effect) and minimize H$_2$O$_2$ inhibition to POD-like activity. For the conventional mixed Fe$_1$/Au NPs system, the low catalytic activity is due to the period required for the transfer of H$_2$O$_2$ from Au NPs to solution and then to Fe SAs. Whereas the low reaction activity of the coplanar Fe$_1$&Au NPs system is ascribed to the inhibition of POD-like activity by the high local concentration of H$_2$O$_2$ generated from glucose oxidization (Supplementary Fig. 26).

As a proof-of-concept application, a colorimetric study on the detection of glucose using the biomimetic Fe$_1$@Au NPs system for enzymatic cascade reactions was performed. As shown in Fig. 4c, the absorbance of oxTMB at 652 nm gradually increases with glucose concentration. The Fe$_1$@Au NPs system based colorimetric biosensor shows a good linear relationship between the absorbance of oxTMB

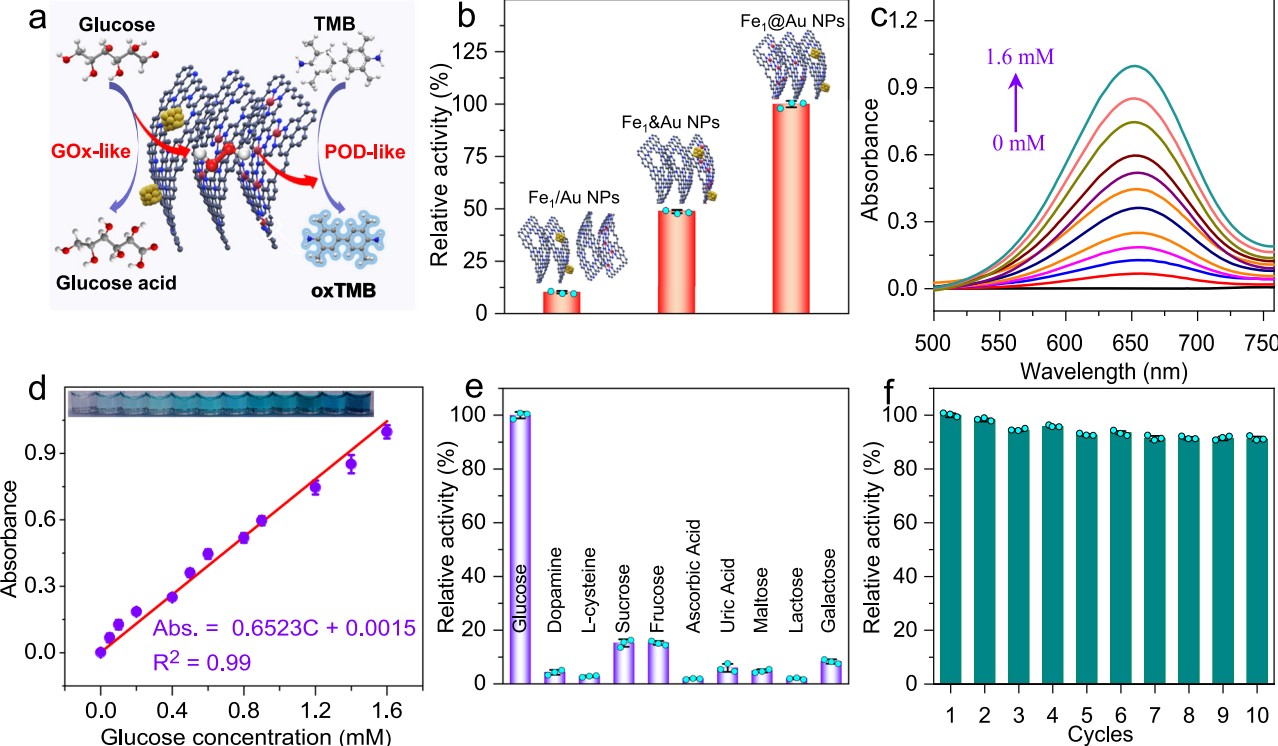

**Fig. 4 | Fe$_1$@Au NPs−based solution sensor for the colorimetric detection of glucose. a** Schematic illustration of colorimetric detection of glucose. **b** Normalized catalytic cascade reaction activities of Fe$_1$/Au NPs, Fe$_1$&Au NPs and Fe$_1$@Au NPs. **c** Absorption spectra of oxTMB with different glucose concentrations. **d** The linear calibration plots for glucose detection. **e** Selectivity evaluation of glucose detection. The concentration of glucose was 0.1 mM, and the concentrations of all interfering substances were 1 mM. **f** Recycling experiments of Fe$_1$@Au NPs-based colorimetric glucose biosensor. Error bars represent standard deviation from three independent measurements.

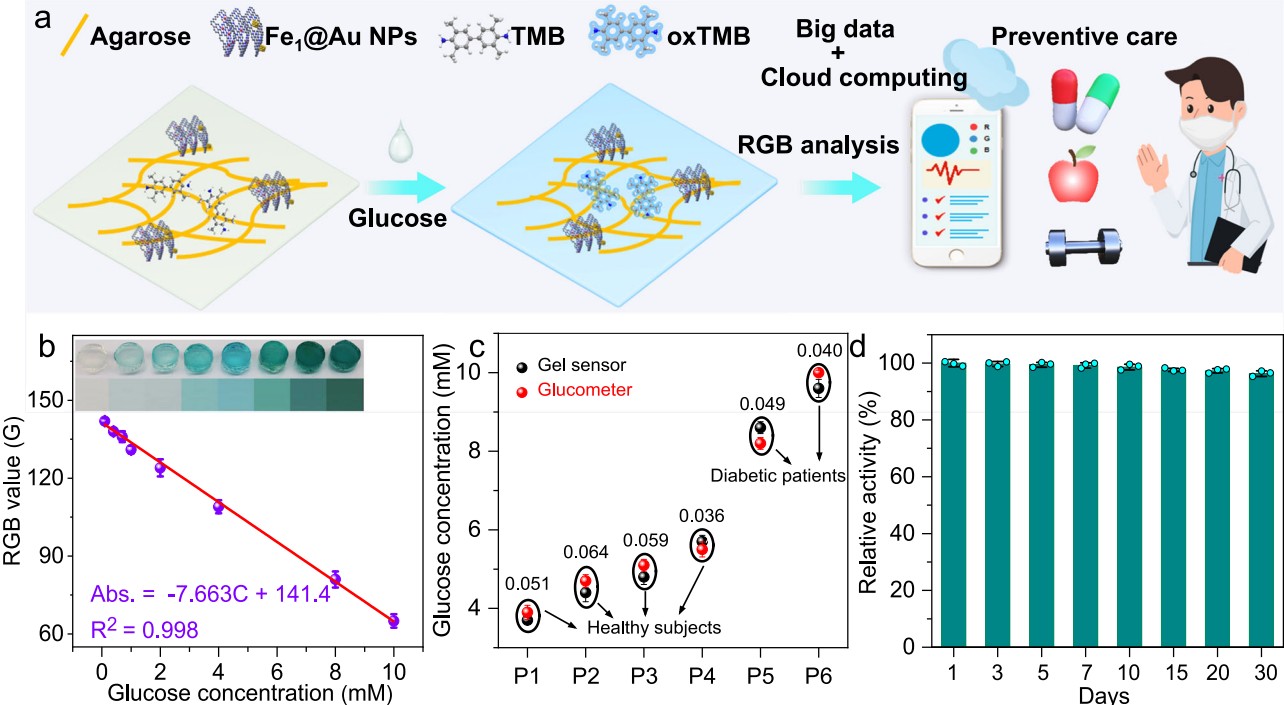

**Fig. 5 | Fe₁@Au NPs–based gel sensor for the colorimetric detection of glucose.**
**a** Schematic illustration of the smartphone-assisted sensing of glucose using all-in-one gel strips. **b** The calibration curves of the RGB value recorded by the smartphone APP toward the different concentration of glucose. **c** Application of gel-based sensor for glucose detection in real serum samples (Note: The glucometer was used as a standard method, and the values near to the circles were a coefficient variation of gel-based sensor for the same sample). **d** The long-term storage stability of gel-based colorimetric sensor. Error bars represent standard deviation from three independent measurements.

and glucose concentrations in the range of 0–1600 µM (Fig. 4d) with a limit of detection (LOD) of 0.13 µM, surpassing most of the reported cascade enzyme-mimicking catalysts-based colorimetric glucose sensors (Supplementary Table 5)[10,30–32]. In addition, the biomimetic Fe₁@Au NPs system displays satisfactory selectivity for glucose sensors in common interfering substances, including dopamine (DA), L-cysteine, sucrose, fructose, ascorbic acid (AA), uric acid (UA), maltose, lactose and galactose (Fig. 4e). Furthermore, the biomimetic Fe₁@Au NPs system exhibits no significant loss of bioactivity for 10 cycle tests, indicating distinguished catalytic stability (Fig.4f and Supplementary Fig. 27). Additionally, no obvious changes in SEM, TEM, HAADF-STEM, and EDS mappings were observed after 10 catalytic cycles, implying a high structural stability of Fe₁@Au NPs (Supplementary Fig. 28). These results reveal that the artificial biomimetic Fe₁@Au NPs system holds great promise for nonenzymatic glucose detection in practical applications.

To further realize the rapid, real-time and visual detection of targets, a portable gel-based colorimetric sensor integrated with a smartphone was constructed for the naked-eye and quantitative analysis of glucose. As illustrated in Fig. 5a, the gel-based sensor was easily established by the immersion of Fe₁@Au NPs and TMB solution into agarose hydrogels. When glucose was added onto the gel strip, the gel colors changed from colorless to blue due to TMB oxidation by using the generated $H_2O_2$ as a mediating substrate. Furthermore, a color recognizer application (APP) named ColorDesk on a smartphone was used to transform colorimetric images into RGB values for quantitative detection. Figure 5b shows that a good linear relationship between the G of RGB values and glucose concentrations was achieved in the range from 0.1 to 10 mM with an LOD of 0.01 mM. Moreover, some real serum samples contributed by volunteers were used to further evaluate the application of the gel-based sensors in disease diagnosis. As shown in Fig. 5c, the measurement results of the gel-based sensor exhibit a high correlation with the levels measured by the commercial glucometer, which can also be verified by favorable coefficients of variation (0.036–0.064). Moreover, the gel-based colorimetric sensor shows good long-term stability, in which 96.3% of its initial response was retained after 30 days (Fig. 5d). In addition, the collected readings can be analyzed with big-data techniques that exhibits great advantages and potentials in point-of-care detection for personalized and ultimately preventive healthcare.

In conclusion, we have reported a cell-inspired design of a biomimetic cascade catalysis system, in which different sites can be spatially integrated into distinct layers of 3D nanocapsules, enabling cascade catalytic reactions with noninterference and high efficiency. The based biomimetic system realizes sensitive detection of glucose by colorimetric sensors with high activity and selectivity owing to proper intermidiate adsorption. Moreover, the proposed gel-based sensors were integrated with a smartphone to realize real-time and visual determination of glucose in real samples and exhibited a high correlation with the commercial glucometer. This work paves a way to designing high-performance biomimetic systems in complex biological environments that hold great promise in a variety of applications.

## Methods
### Chemicals
All chemicals were used as received without further purification. Ferric chloride hexahydrate (FeCl₃.6H₂O), 5, 10, 15, 20-tetra (4-(imidazol-1-yl) phenyl) porphyrindine (TIPP), polyvinylpyrrolidone (PVP, Mw~4000) and tetraethyl orthosilicate (TEOS) were purchased from Aladdin Company. Ethanol (CH₃CH₂OH, 99.7%), N, N-dimethylformamide (DMF, AR), α,α'-Dibromo-p-xylene, sodium sulfate anhydrous (Na₂SO₄), sodium hydroxide (NaOH), hydrogen chloride (HCl, 37%) were purchased from Sinopharm Chemical Reagent Co., Ltd. Chloroauric acid (HAuCl₄·4H₂O) was purchased from Sigma-Aldrich. Dopamine HCl was purchased from Beijing HWRK Chem Co., Ltd. 3,3',5,5'-tetramethylbenzidine (TMB) and PBS were gotten from Sangon

Biotech (Shanghai) Co., Ltd. Glucose, fructose, sucrose, L-cysteine, ascorbic acid, dopamine, uric acid, and maltose were purchased from Beijing Chemical Reagent Company. Deionized (DI) water from Milli-Q System (18.2 MΩ·cm, Millipore, Billerica, MA) was used in all experiments.

## Synthesis of peanut-shaped $Fe_2O_3$

In a typical synthesis, $FeCl_3 \cdot 6H_2O$ solution (50 mL, 2 M) was stirred in an oil bath at 75 °C for 5 min. Then, NaOH solution (50 mL, 5.4 M) was dropwise added the above solution. After stirring for 15 min, $Na_2SO_4$ solution (50 mL, 0.6 M) was introduced into the above-mixed solution. Finally, the obtained $Fe(OH)_3$ gel was transferred to 100 mL Teflon-lined stainless-steel autoclaves and then heated at 100 °C for 5 days. The red product was washed via centrifugation with ethanol and deionized (DI) water for three times. The obtained crystals were dried under vacuum at 80 °C.

## Synthesis of $Fe_2O_3$@PDA

The pre-synthesized $Fe_2O_3$ (350 mg) was dispersed by ultrasound in 400 mL of freshly prepared Tris-buffer solution (10 mM, pH 8.5). Then dopamine-HCl (160 mg) was added to the above buffer solution. The mixed solution was allowed to stir for 2 h at room temperature. The resulting product was washed with deionized water and ethanol for three times, and collected by centrifugation. After dried at 80 °C in a vacuum, the desired $Fe_2O_3$@PDA was obtained.

## Synthesis of peanut-shaped $Fe_1$/NC

In a typical procedure, the $Fe_2O_3$@PDA power was transferred into a ceramic boat and placed in a tube furnace. The sample was annealed at 500 °C for 2 h and then 700 °C for 1 h under Ar atmosphere with a heating rate of 5 °C min$^{-1}$. Next, the metal oxides were removed by immersing the samples in the 5 M HCl solution for 6 h at 80 °C. Finally, the obtained powder was thoroughly washed by using deionized water and then dried in a vacuum at 80 °C.

## Synthesis of $Fe_1$@Au NPs

The $HAuCl_4$ solution (200 μL, 10 mM) was added into the $Fe_1$/NC aqueous solution (20 mL, 0.1 mg mL$^{-1}$). Subsequently, the mixture was stirred for 1 min, followed by the addition the ice-cold, freshly prepared $NaBH_4$ aqueous solution (50 μL, 0.1 M). Then the mixture was immediately washed by centrifuging with water for twice. Finally, the obtained $Fe_1$@Au NPs was freeze-dried for further use.

## Synthesis of $Fe_1$&NC

The $SiO_2$ spheres were initially synthesized employing classical Stöber method. A mixture, comprising 150 ml of ethanol, 50 ml of deionized water, and 7.5 ml of 28% $NH_3 \cdot H_2O$, was stirred at ambient temperature. Then, 6 ml of tetraethyl orthosilicate was swiftly introduced into the aforementioned mixture. Following approximately 1 h of reaction time, the $SiO_2$ spheres were harvested via centrifugation. Subsequently, 80 mg of α,α′-dibromo-p-xylene was dissolved in 40 ml of DMF, resulting in a lucid solution. This solution was then amalgamated with another solution containing 40 mg of as-prepared Fe-TIPP, 20 mg of TIPP, and 80 mg of $SiO_2$ under vigorous agitation. The resulting mixture was stirred in an oil bath at 110 °C for 24 h. The resultant product was washed with DMF and ethanol through centrifugation, and thereafter desiccated at 80 °C under vacuum overnight. Subsequently, the obtained powder was translocated into a tube furnace and annealed at a temperature of 800 °C for 3 h, utilizing $H_2$/Ar gases, with a heating rate of 5 °C min$^{-1}$. Finally, the accomplished product was etched in a 5 M NaOH solution for a duration of 24 h, effectuating the removal of $SiO_2$, and thereby obtaining the Fe single atoms anchored on the surface of NC shell (denoted as $Fe_1$&NC).

## Synthesis of $Fe_1$&Au NPs

The synthesis process of $Fe_1$&Au NP is similar with $Fe_1$@Au NPs, where $Fe_1$&NC aqueous solution (20 mL, 0.1 mg mL$^{-1}$) was used instead of $Fe_1$/NC aqueous solution.

## Synthesis of Au NPs

Gold nanoparticle was prepared according to the reduction of gold (III) complex sodium citrate. A 50 mL bright yellow $HAuCl_4$ solution (0.12 mM) was heated until 80 °C in a round-bottom flask, followed by the dropwise addition of 1 mL of sodium citrate (85 mM). The above-mixed solution was kept under heat for 1 h, and the solution became black, and then changed to deep red. Then 200 μL as-prepared gold nanoparticle was diluted into 2.0 mL with deionized water. The 10 mg PVP (MW: 40000) was subsequently added. After 6 h stirring, the PVP modified gold nanoparticle was centrifuged for 10 min and washed with deionized water for three times.

## Characterization

TEM images were acquired utilizing a Hitachi-7650 microscope, operating at an acceleration voltage of 100 kV. For high-resolution TEM images and corresponding EDS analysis, a JEOL JEM-2100F microscope was employed, operating at 200 kV. HAADF-STEM images were captured using a FEI Titan Cubed Themis G2 300 equipped with a probe corrector, also operating at 200 kV. XRD measurements were performed employing a Rigaku Miniflex-600 instrument. Raman shifts were meticulously analyzed using a cutting-edge LabRAM Aramis Raman spectrometer (Solid 3700). XPS spectra were acquired at VG-Multilab2000 instrument. FT-IR spectra were obtained using a Nicolet 8700 FT-IR instrument. Elemental analysis of Fe and Au in the samples was performed using inductively coupled plasma atomic emission spectrometry (Optima 7300 DV). Nitrogen sorption measurement was carried out using a Micromeritics ASAP 2020 system. The pore size distribution was calculated using the HK and BJH methods for micropore and mesoporous, respectively.

Soft X-ray absorption spectra (N-Kedge) were acquired at the BL12B X-ray Magnetic Circular Dichroism station and BL10B photo-emission end-station of the National Synchrotron Radiation Laboratory in Hefei, China. XAFS data (Fe K-edge) was obtained at the 1W1B station in the Beijing Synchrotron Radiation Facility (BSRF). The BSRF storage rings operated at an energy of 32.5 GeV with a maximum current of 250 mA. The XAFS data was recorded in transmission mode, employing an $N_2$-filled ionization chamber. The acquired data was processed using the ATHENA module in the IFEFFIT software packages. The $k^3$-weighted EXAFS spectra were obtained by subtracting the post-edge background from the overall absorption and normalizing with respect to the edge-jump step. The samples were pelletized as 13 mm diameter disks using graphite powder as a binder. The $k^3$-weighted χ (k) data in k-space was Fourier transformed to real (R) space using a Hanning window (dk = 1.0 Å$^{-1}$) to separate the EXAFS contributions from different coordination shells.

## Steady-state kinetic assays of the POD-mimicking activity

Kinetic experiments were monitored in a reaction volume of 200 μL HAc-NaAc buffer solution (pH = 4.0) containing 20 μg mL$^{-1}$ the POD-mimicking catalyst, TMB (0.1–1.2 mM) as a substrate and $H_2O_2$ (6 mM), or $H_2O_2$ (0.1–10 mM) as a substrate and TMB (0.6 mM). The mixture solution was incubated at room temperature for 10 min and then used for UV-vis absorbance measurement at 652 nm using a THERMO Varioskan Flash spectrophotometer. The Michaelis-Menten constant was calculated according to the Michaelis-Menten equation:

$$V_0 = \frac{V_{max}[S]}{(K_m + [S])} \tag{1}$$

The $V_0$ is the initial velocity, $V_{max}$ corresponds to the maximum reaction velocity, which is monitored when the catalytic sites on the POD mimics are saturated with substrate. [S] is the initial substrate concentration, and Km is the Michaelis constant. The initial velocity $V_0$ was determined by according the slope of the kinetic curve in the initial phase, and the substrate concentration [S] was obtained at t = 0 s. The kinetic parameters Km and $V_{max}$ were fitted according to Michaelis–Menten equation based on the calculated $V_0$ and [S].

## Steady-state kinetic assays of the GOx-mimicking activity

The kinetic behavior of Fe$_1$@Au NPs was studied by monitoring the absorbance in 30 s intervals while varying the glucose concentration. The Michaelis–Menten constant was calculated using Lineweaver-Burk plots of the double reciprocal based on the above-mentioned Michaelis–Menten equationt.

## Characterization of GOx-like activity

The Fe$_1$@Au NPs (20 µL, 200 µg/ml), and glucose (20 µL, 10 mM) were added into 96-well plates containing PBS buffer (160 µL, 10 mM, pH 7.2). Then the mixed solution was incubated for 30 min at 37 °C. Next, the mixture was centrifuged for 3 min to obtain the supernatant. The generated $H_2O_2$ in the supernatant (60 µL) was verified by the HRP (20 µL, 50 µg/mL) - TMB (20 µL, 6 mM) based chromogenic reaction in HAc-NaAc buffer (100 µL, 0.1 M, pH = 4.0). The absorbance spectra were recorded by using a THERMO Varioskan Flash spectrophotometer. Gluconic acid as another product was verified by a specific colorimetric assay. Briefly, the obtained gluconic acid supernatant (100 µL) was added into solution 1 (250 µL, 5 mM EDTA and 0.15 mM trimethylamine in water), and then the mixed solution was added into solution 2 (25 µL, 3 M NH$_2$OH in water). After incubation for 25 min, the solution 3 (125 µL, 1 M HCl, 0.1 M FeCl$_3$ and 0.25 M CCl$_3$COOH in water) was added to the aforementioned solution, and the mixture was incubated for 10 min before spectral measurement.

## Characterization of peroxidase-like activity

The Fe$_1$@Au NPs (20 µL, 200 µg/ml), $H_2O_2$ (20 µL, 10 mM), and TMB (20 µL, 6 mM) were added into 96-well plates containing PBS buffer (140 µL, 10 mM). Then, the mixed solution was incubated for 10 min before spectral measurement.

## Specific activity of the enzyme-mimicking catalysts

The specific activity (SA), which is defined as activity units per milligram of the enzyme-mimicking catalysts. The activity (units) of the enzyme-mimicking catalysts was calculated using Eq. (2):

$$b_{enzyme\ mimics} = \frac{V \times (\frac{\Delta A}{\Delta t})}{\varepsilon \times 1} \tag{2}$$

$B_{enzyme\ mimics}$ is the enzyme-mimicking catalytic activity of the Fe$_1$@AuNPs expressed in units. V is the total volume of the reaction solution (µL); $\varepsilon$ is the molar absorption coefficient of the colorimetric TMB (39,000 M$^{-1}$ cm$^{-1}$); l is the path length of light traveling in the cuvette (cm); A is the absorbance value; and $\Delta A/\Delta t$ is the initial rate of change in absorbance at 652 nm min$^{-1}$. The SA of the enzyme-mimicking catalysts (U mg$^{-1}$) is calculated in single active sites: $a_{enzyme\ mimcs} = b_{enzyme\ mimcs}/m$. Where $a_{enzyme\ mimcs}$ is the SA expressed in units per milligram (U mg$^{-1}$) enzyme-mimicking catalysts, and m is the enzyme-mimicking catalyst weight (mg) of each assay.

## Electron paramagnetic resonance (EPR) experiments

EPR measurements were carried out by using the JES-FA200 system. 5,5-dimethyl-1-pyrroline Noxide (DMPO) was used as the spin trapping agent to capture active species (·OH) in the reaction. The same quartz capillary tube was used to minimize experimental errors in all EPR measurements. In a normal measurement, the catalyst (20 µL, 20 µg/mL) was added to a mixture of TMB (50 µL, 1.2 mM), $H_2O_2$ (20 µL, 1 M) and DMPO (20 µL) in 1 mL buffer solution. EPR spectrum was recorded after 1 min of reaction.

## Verification of Intermediate (•OH)

The blue methylene blue (MB) could be degraded to the colorless products in the presence of •OH. Therefore, MB is usually employed to verify the existence of •OH by colorimetric assay. The catalyst (1 mg mL$^{-1}$, 100 µL) was added into the buffer solution (1 mL, 0.1 M) containing $H_2O_2$ (1 M, 1 mL) and MB (1 mM, 100 µL). Then, the absorbance of the reaction solution was monitored after 1.5 h.

## Characterization of biomimetic cascade catalysis

The Fe$_1$@Au NPs (20 µL, 200 µg/ml), and glucose (20 µL, 10 mM) were added into 96-well plates containing PBS buffer (140 µL, 10 mM, pH 7.2). Then the mixed solution was incubated for 30 minutes at 37 °C. Next, the TMB (20 µL, 6 mM) was added to the above solution. Finally, the mixture was incubated for 10 min before spectral measurement.

## Colorimetric biosensor of glucose

The Fe$_1$@Au NPs (20 µL, 200 µg/ml) and glucose (20 µL) with different concentrations were added into 96-well plates containing PBS buffer (140 µL, 10 mM, pH 7.2). Then the mixed solution was incubated for 30 min at 37 °C. Next, the TMB (20 µL, 6 mM) was introduced to the above solution. Finally, the mixture was incubated for 10 min and detected at 652 nm using a THERMO Varioskan Flash spectrophotometer.

## The selectivity evaluation of glucose detection

The selectivity of Fe$_1$@Au NPs system for glucose was detected in solution containing glucose (0.1 mM) or interfering substances (dopamine (DA), L-cysteine, sucrose, fructose, ascorbic acid (AA), uric acid (UA), maltose, lactose and galactose, 1 mM). Specifically, the Fe$_1$@Au NPs (20 µL, 200 µg/ml) and glucose (20 µL, 1 mM) or other interfering chemicals (20 µL, 10 mM) were added into 96-well plates containing PBS buffer (140 µL, 10 mM, pH 7.2). Then, the mixed solution was incubated for 30 min at 37 °C. Next, the TMB (20 µL, 6 mM) was introduced to the above solution. Finally, the mixture was incubated for 10 min and absorbance was detected at 652 nm using a THERMO Varioskan Flash spectrophotometer.

## Recycling experiments

In the recycling experiments, the Fe$_1$@Au NPs (20 µL, 200 µg/ml) and glucose (20 µL, 10 mM) were added into a tube containing PBS buffer (140 µL, 10 mM, pH 7.2). The mixed solution was incubated for 30 min at 37 °C. Then, the TMB (20 µL, 6 mM) was introduced to the above solution. The mixture was incubated for 10 min and detected at 652 nm using a THERMO Varioskan Flash spectrophotometer. Next, the supernatant was removed after the test, and a new portion of glucose and PBS buffer and TMB were added. The mixture was incubated and detected at 652 nm using a THERMO Varioskan Flash spectrophotometer. Recycling continued for ten runs. All reactions were duplicated.

## Storage stability tests

The storage condition of Fe$_1$@Au NPs-based sensor was at 4 °C temperature in dark. The long-term stability of Fe$_1$@Au NPs based glucose biosensor was evaluated by measuring the response towards the same glucose concentrations for every few days. Specifically, the Fe$_1$@Au NPs (20 µL, 200 µg/ml) and glucose (20 µL, 10 mM) were added into 96-well plates containing PBS buffer (140 µL, 10 mM, pH 7.2). Then the mixed solution was incubated for 30 min at 37 °C. Next, the TMB (20 µL, 6 mM) was introduced to the above solution. Finally, the mixture was incubated for 10 min and detected at 652 nm using a THERMO Varioskan Flash spectrophotometer.

## Preparation of the integrated agarose-based gel film

Agarose hydrogels of 1.0% (w/v) were prepared in a tube by dissolving 0.2 g of agarose in 20 mL 1×PBS solution followed by microwaving 60 s to complete the solubility of agarose. A thermometer was inserted into the above solution to monitor its temperature. When the temperature of the gel cooled to about 40 °C, TMB solution (1 mL, 6 mM), $Fe_1$@Au NPs (200 μL, 1 mg/mL, these solutions were preheated to 37 °C) were quickly added into the above solution in sequence with properly stirring. The film casting was done by pouring the mixed solution into a petri dish and cooling to room temperature, and then the gel was divided into small wafers (R = 0.6 cm) for use.

## Gel-based sensor for glucose detection

The glucose with different concentrations (20 μL, 0-10 mM) were introduced dropwise on the gel strips. Following a 30-minute incubation period, the gel strips were placed into a homemade colorimetric box for RGB color analysis using a smartphone. The entire procedure was executed under a uniformly internal illumination with a consistent size and layout, thereby ensuring that ambient light exerted no influence on the color measurements. The ColorDesk app, a portable color digitizer tool, was used to obtain RGB value of the gel strips in real-time. In our study, we selected the RGB color space mode, which represents the combination of red (R), green (G), and blue (B) color components.

## Data processing of gel-based sensor

In terms of data processing, the principle of detection was as follows: the color intensity increased as the assay solution color became darker, which depended on the glucose concentration. According to the RGB color space, any color can be disentangled into R, G, and B components. The intensity values for each RGB channel range from 0 to 255, with higher values indicating brighter colors. A value of zero corresponds to the strongest intensity (dark color), while a value of 255 represents the lowest intensity (white color). Therefore, a solution with an intense color (high glucose concentration) has low RGB values, and vice versa. To determine the most suitable relationship for quantifying a scanned image, different quantitative relationships, including R, G, and B, were analyzed. Among all the relationships studied, G showed a strong correlation with the glucose concentration. Therefore, the intensity of G was selected as the analytical signal for detection.

## Human subjects and serum samples analysis

Human serum samples were collected from diabetic patients and healthy subjects with informed consent at the First Affiliated Hospital of University of Science and Technology of China (USTC). Six human blood samples (including 4 males and 2 females based on self-report, all human subjects signed informed consent and received 200 RMB in compensation) with unknown glucose concentrations were analyzed by the same process in the section of "Gel-based sensor for glucose detection" just by replacing glucose solutions with the human serum samples. For comparison, the glucose levels in serum samples were also measured by a commercial glucometer (Abbott, FreeStyle Optium Neo, Mexico) and the blood glucose test strips (Abbott, FreeStyle Optium Neo, U.K.). All procedures are approved by Institutional Ethics Review Committee of the First Affiliated Hospital of USTC (2021 KY 089).

## Reporting summary

Further information on research design is available in the Nature Portfolio Reporting Summary linked to this article.

## Data availability

The data supporting the findings of this study are available within the article and its Supplementary Information files (Supplementary Information, Supplementary Data 1). All other relevant source data are available from the corresponding authors upon request.

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

## Acknowledgements

This work was supported by China Ministry of Science and Technology 2021YFA1500404 (C.X.), the National Natural Science Foundation of China [92261105 (Y.W.), 22221003 (Y.W.) and 22201271(H.Z.)], the Anhui Provincial Natural Science Foundation [2108085QB70 (H.Z.) and 2108085UD06 (Y.W.)], the Key Technologies R & D Program of Anhui Province (2022a05020053, H.Z.), the Collaborative Innovation Program of Hefei Science Center, CAS (2021HSC-CIP002, Y.W.), the Natural Science Foundation of Hefei, China (Grant No. 2021044, Y.W.), the Strategic Priority Research Program of the Chinese Academy of Sciences (XDA21061009, Y.W.), the Fundamental Research Funds for the Central Universities (Grant WK2060190103, Y.W.), the Joint Funds from Hefei National Synchrotron Radiation Laboratory [KY2060000180 (H.Z.) and KY2060000195 (C.X.)], and the China Postdoctoral Science Foundation funded project (2021TQ0216, X.G.). This work was partially carried out at the USTC Center for Micro and Nanoscale Research and Fabrication. Thank you for the funding support from the CAS Fujian Institute of Innovation. We acknowledge the Experimental Center of Engineering and Material Science at the University of Science and Technology of China. We thank the photoemission end stations BL1W1B in Beijing Synchrotron Radiation Facility (BSRF), BL14W1 in Shanghai Synchrotron Radiation Facility (SSRF), BL10B and BL11U in National Synchrotron Radiation Laboratory (NSRL) for the help in characterizations. The DFT calculations in this work were performed at the Supercomputing Center of the University of Science and Technology of China.

## Author contributions

Y.W. and H.Z. conceived the experiments and designed the study and wrote the paper. Q.W. and K.C. planned synthesis, carried out characterization and catalytic measurement and wrote the paper. J.X. performed the electron-microscopy characterization. X.G. performed the DFT calculations. C.C. drew the pictures. S.X. assists in study design and proofreads the manuscript. H.J., S.X., C.X. and M.C. discussed the results and helped with the modification of the paper.

## Competing interests

The authors declare no competing interests.
