## [Peer review file · Nature Communications]

Reviewers' comments:

Reviewer #1 (Remarks to the Author):

This article deals with a system that mimics the behaviors of cells and able to detect selectively specific molecules of health and biological interest, such as glucose. The submitted work, consider glucose as target species. In general, the work is of interest. However, it needs improvement as the manuscript has linguistic problems and lacks of a coherent and logical organization in the presentation and discussion of the obtained results.

In many places, the manuscript is very difficult to read. In addition, the authors should avoid repeating same concepts throughout the text.

As for the novelties, the authors should mention in the introduction section, which is the advantage of their proposed approach/system with respect to similar strategies/systems available in the literature, where cascade reactions, mimicking cell behaviors, have been exploited for the detection of glucose. To this end, the authors can consider, for instance, a recent review on metal-organic frameworks (Adv. Funct. Mater.2021, 31, 2106023) which have recently been used for the construction of a variety of electrochemical and optical glucose sensors.

Specific points.

Abstract. It does not provide suitable information on the manuscript's content. It needs rewriting.

Lines 76-78. The incipit of the Results and Discussion section actually repeats the same concept already stated in the introduction, concerning the compartmentalized activity of an enzyme. Avoid it and start by describing the results obtained in the investigation.

Line 84. Please, use the acronym PDA in parenthesis after polydopamine.

Figure-S10. The experimental traces (dashed lines) are not visible.

Line 156. This sentence is confusing and misleading. For what species /compound is the colorimetric assay employed?

Lines 167-162. Here, the authors do not provide enough information on the measurements performed to show the POD-mimicking ability of the Fe1@Au NPs system. Here, results are given and commented in Figures S12-14. However, captions to the figures do not explain the various experiments done. In particular, apparently, the authors employed different compounds with different combinations of Fe1 and AuNPs, but, at this stage, the reader cannot understand how the latter materials differ from Fe1@Au NPs. Afterwards, later on, lines 166-177, they comment the meaning of the various acronyms and summarize relevant results in Figure 3. The reader gets really confused! The results and the related discussion need to be better organized.

Line 164. The authors should comment what advantages present their system with respect to those reported in Refs. 21-23, and other systems recently considered in the above mentioned review (i.e., Adv. Funct. Mater.2021, 31, 2106023).

Lines 165 – 169. These sentences express the concept of cascade catalysis, repeatedly stated by the authors. Please avoid it, and just go to the further results obtained.

Line 197. " The based biosensor..". What does it mean?

Lines 203-206. SEM, TEM and other surface techniques cannot provide information on "bioactivity".

Line 209. It is not clear to me what do the authors mean with "To further realize the instrument-

free,...(?)

Lines 213-217. It is not clear how the gel filled with Fe₁@Au NPs works. Specifically, how did the authors include glucose in the gel? How does the color recognition application work? How long does it take from the addition of glucose to give the optical response? How stable is the sensor? This information should be provided not only for the gel/Fe₁@Au NPs system, but also generally for any material employed in the manuscript.

Lines 222 – 245. DFT results should be presented and discussed earlier.

Reviewer #2 (Remarks to the Author):

The construction of artificial biomimetic systems to ensure cascade reactions with non-interfering and high efficiency, as in living cellular systems, is a great challenge. In this manuscript, Wu et al. ingeniously developed a cell-inspired design of a cascade catalysis system with highly spatially separated sites, where sites can catalyze independently and work synergistically to realize highly efficient cascade glucose detection. The novel construction method solves the interference between reactions in cascade catalysis, which would greatly drive the development of biomimetic catalysis. In practice, the performance of the biomimetic system is fairly good in both colorimetric and gel-based glucose sensors. In addition, the nanostructure of the nanocapsules was systematically characterized by state-of-the-art characterization methods, and the catalytic reaction and its mechanism were further studied by DFT calculations. In summary, this is interesting and ground-breaking work. This work is recommended to be published in Nature Communications after minor revision. However, some critical issues should be addressed.

1. In this work, the spatial separation between Fe SAs and Au NPs is important for highly efficient cascade catalysis. How do the authors achieve the highly spatially separated distribution of these different sites?
2. Why do authors select Fe SAs as POD mimics, instead of other metal SAs (such as Cu, Co, Ni, ...)? Is there anything special advantage about Fe SAs?
3. In cascade reactions, the integration of different kinds of nanozymes in one system would accompany a low cascade reaction efficiency. Why, in this work, the biomimetic cascade catalysis system is so efficient?
4. In page 4, the authors referred "Acid leaching was further introduced to remove the remained Fe₂O₃ ...", however, "Acid leaching" had not be stated in the whole text. Besides, Fe₂O₃ was removed by acid leaching, but why Fe SAs can be remained? The relative statement about the detailed changes of the sample in this process should be supplied.
5. In Figure 2a, there is a broad peak in Fe₁/NC, but the explanation is lost.
6. The EPR spectra presented in Figure S15 probably have different line widths, which may call into question assertions about the relative catalytic activity, unless the intensities were obtained by careful integration and baseline corrections.
7. I noticed that the authors use the APP to realize real-time and visual determination of glucose displayed in Figure 3g, and relate RGB values and concentration of glucose, what's the process? Also, how to get the RGB values by the APP?

Reviewer #3 (Remarks to the Author):

This manuscript reports Cell-inspired nanozyme for biomimetic cascade catalysis toward glucose and its application for glucose detection. The results and experimental data appear to be satisfactory. However, there are certain issues that must be resolved to improve the quality of the manuscript. The followings are the suggestions for improving its quality.

1. There is no experimental details for selectivity test in Fig S22. How much concentration of glucose and interfering substances? What experimental method was used to get and evaluate the activity toward glucose and substances? In addition, galactose and lactose (highly possible interfering substances) should be investigated for selectivity test. Note that the applied concentration of interfering substances for selectivity should be based on their physiological level.
2. Give the experimental details for Fig S23.
3. The authors described that "When glucose was added onto the gel strip, the gel colors changed from colorless to blue due to TMB oxidation by the using the generated H₂O₂ as a mediate substrate.". The resulting blue color was demonstrated in Fig 3(f) and (h). However, contradictively, authors chose Green value of RGB for quantitative analysis instead of Blue value. Experiments and data analysis involved in colorimetric assay are unclear and vague.
4. What kind of color recognizer application (APP) did you use? Did you develop it? The authors should provide experimental details unfeignedly.
5. The interference from colored erythrocytes (red blood cells) present in whole blood would seriously affect the interpretation results. The authors have the solution to deal with this problem? The authors need to add the discussion on it.
6. The authors should give the real glucose test (e.g. human serum or blood) and the comparison with glucometer to demonstrate the feasibility of gel-based colorimetric sensing system.
7. Aside from the stability of Fe₁@Au NPs shown in Fig S23, the long-term stability of gel-based colorimetric sensor should be investigated for practical applications.

Response to referees and revisions made in the manuscript:

We thank the reviewers for their helpful and constructive comments and suggestions. We have responded to each and every question as detailed below.

Reviewer #1: This article deals with a system that mimics the behaviors of cells and able to detect selectively specific molecules of health and biological interest, such as glucose. The submitted work, consider glucose as target species. In general, the work is of interest. However, it needs improvement as the manuscript has linguistic problems and lacks of a coherent and logical organization in the presentation and discussion of the obtained results.

In many places, the manuscript is very difficult to read. In addition, the authors should avoid repeating same concepts throughout the text.

As for the novelties, the authors should mention in the introduction section, which is the advantage of their proposed approach/system with respect to similar strategies/systems available in the literature, where cascade reactions, mimicking cell behaviors, have been exploited for the detection of glucose. To this end, the authors can consider, for instance, a recent review on metal-organic frameworks (Adv. Funct. Mater.2021, 31, 2106023) which have recently been used for the construction of a variety of electrochemical and optical glucose sensors.

Response: We are grateful to the Reviewer #1 for his/her recognition and appreciation of our work. We took the reviewers' comments carefully. As suggested, the native English speakers have carefully checked every sentence and corrected grammatical issues in our manuscript. For example, "cost" in page 2 was corrected as "costs". "show" in page 3 was corrected as "shows". "constructed successfully" in page 4 was corrected as "successfully constructed". "efficient" in page 6 was corrected as "efficiently". "design" in page 10 was corrected as "designing". In addition, the whole manuscript was polished by Nature Author Services to ensure linguistic issues resolved. The revised parts are highlighted in yellow in both the revised manuscript and supporting information.

Furthermore, we reorganized the results and the related discussions of Fig.3, and revised unclear sentences in original manuscript. For example, "the generations were verified by a specific colorimetric assay" was revised to "the generated gluconic acid and H₂O₂ were verified by a specific colorimetric assay". "The based sensor" was revised to "The Fe₁@Au NPs system based colorimetric biosensor".

Additionally, we deleted repeated concepts about the compartmentalized activity of an enzyme and cascade catalysis in our revised manuscript, and added the novelties of the proposed system on pages 2-3 and 8 with respect to similar systems. Besides, the above-mentioned literature is included in the References list (reference [13]).

Overall, we have answered all questions from the Reviewer #1. The specific replies and modifications are listed as follows:

Comment 1: Abstract. It does not provide suitable information on the manuscript's

content. It needs rewriting.

Response 1: We thank the reviewer for the careful reading of our manuscript and the enlightening suggestion. As suggested, the abstract has been rewritten to state suitable information on the manuscript's content.

Abstract: Individual cells contain multiple isolated compartments that enable the spatial confinement of enzymes in different cellular domains, allowing enzymatic cascade reactions to occur with high efficiency. Herein, we report a cell-inspired design of a biomimetic cascade catalysis system by immobilizing Fe single atoms and Au nanoparticles on the inner and outer layers of three-dimensional nanocapsules, respectively, **in which Fe single atoms serve as peroxidase mimics and Au nanoparticles act as artificial glucose oxidases.** The different metal sites catalyze independently and work synergistically to enable engineered and cascade glucose detection. The obtained biomimetic catalysis system demonstrates ~ 9.8- and 2-fold cascade activity enhancement than systems with metal sites prepared by conventional mixing and coplanar construction, respectively. **Furthermore, the biomimetic catalysis system is successfully demonstrated for the colorimetric detection of glucose with a high sensitivity (detection limit as low as 0.13 μ M), high catalytic activity and selectivity.** Also, the proposed gel-based sensor is integrated with a smartphone to enable real-time and visual determination of glucose. **More importantly, the gel-based sensor is successfully applied for glucose detection in real samples and exhibits a high correlation with a commercial glucometer. These findings provide a strategy to design an efficient biomimetic catalysis system for applications in bioassays and nanobiomedicines.**

The "Abstract" section was shown in page 1-2 of the revised manuscript.

Comment 2: Lines 76-78. The incipit of the Results and Discussion section actually repeats the same concept already stated in the introduction, concerning the compartmentalized activity of an enzyme. Avoid it and start by describing the results obtained in the investigation.

Response 2: We thank the reviewer for this valuable suggestion. The repeated concept about the compartmentalized activity of an enzyme has been deleted in our revised manuscript.

Comment 3: Line 84. Please, use the acronym PDA in parenthesis after polydopamine.

Response 3: Thanks for the good comment. As suggested, we have added the acronym PDA after polydopamine in line 89 (page 3) of the revised manuscript.

Comment 4: Figure-S10. The experimental traces (dashed lines) are not visible.

Response 4: Thanks for the good comment. We are very sorry that the figures presented were not clear. To better evaluate the quality and interpretation of the figures, these figures have been revised. We have checked carefully to ensure that it will read in a clear format.

The revised images (Fig.R1) were showed in supporting information as Fig.S10 (in the page S21).

Fig. R1 EXAFS fitting curves of Fe₁@Au NPs at (a) R space and (b) k-space.

Comment 5: Line 156. This sentence is confusing and misleading. For what species /compound is the colorimetric assay employed?

Response 5: We are very sorry for not giving the specific species/compound in this sentence, which may cause difficulties for readers to understand our work. The species/compound refer to gluconic acid and H₂O₂.

The revised sentence of “the generated gluconic acid and H₂O₂ were verified by a specific colorimetric assay” was showed in line 162-163 of page 6.

Comment 6: Lines 167-162. Here, the authors do not provide enough information on the measurements performed to show the POD-mimicking ability of the Fe₁@Au NPs system. Here, results are given and commented in Figures S12-14. However, captions to the figures do not explain the various experiments done. In particular, apparently, the authors employed different compounds with different combinations of Fe₁ and AuNPs, but, at this stage, the reader cannot understand how the latter materials differ from Fe₁@Au NPs. Afterwards, later on, lines 166-177, they comment the meaning of the various acronyms and summarize relevant results in Figure 3. The reader gets really confused! The results and the related discussion need to be better organized.

Response 6: We are very sorry for our unclearly explanation, which may cause difficulties for readers to understand our work. To further verify the POD-mimicking activity of Fe₁@Au NPs, *o*-phenylenediamine (OPD) and 2, 2'-azino-bis(3-ethylbenzothiazoline-6-sulfonic acid) (ABTS) were also selected as the chromogenic substrates. As shown in Fig.R2, Fe₁@Au NPs can catalyze the oxidation OPD to give an orange color and ABTS to give a green color in the presence of H₂O₂. These results all indicate the POD-mimicking activity of Fe₁@Au NPs.

We are very sorry for missing the definitions of SA, Fe₁/Au NPs and Fe₁&Au NPs shown in Figure 3, which may cause confusion for readers. The definitions and descriptions of various acronyms were added in revised manuscript.

“The conventional mixing system was obtained by physically mixing Fe SAs with Au NPs sites (denoted as Fe₁/Au NPs). The coplanar construction system was prepared according to the previous reports (denoted as Fe₁&Au NPs, both Fe SAs and Au NPs sites are on the surface of the support).” The related description about the compounds with different combinations of Fe SAs and Au NPs has been advanced to line 175-179 of page 6 in order to better evaluate the quality and interpretation of the manuscript.

The captions to the Figures S12-14 have been revised to explain accurately the various experiments done. For example, “the characteristic absorption peak of oxTMB with a clear blue color was found, whereas individual Fe₁@Au NPs/TMB and H₂O₂/TMB cannot lead to a color change” in Figure S12 was revised as “Fe₁@Au NPs can catalyze the oxidation of TMB to generate the blue oxTMB **with the distinct absorption peak at 652 nm. Negligible absorbance in the UV-Vis absorption spectra was observed in Fe₁@Au NPs/TMB and H₂O₂/TMB**”. In addition, **optical image of the reaction system was added** in Figure S12. Besides, Figure S13a and S13b were switched, and “4.0 and 30°C” was corrected as “5.0 and 30°C”. The caption to the Figures S14 was transferred to the manuscript as shown in line 179-188. The corresponding image (Figure S14) was transferred to the revised manuscript as Fig.3b-3c and supporting information as Fig.S17.

Besides, the results of Fig.3 have been revised and reorganized as shown in Fig.R3–R6.

The related description and discussion were showed in line165-188 (page 6-7) of the revised manuscript and page S23-24 of the supporting information. The corresponding images (Fig.R2-R6) were shown in revised manuscript as Fig.4-5 (page 17-18) and supporting information as Fig.S12 (page S23), Fig.S23 (page S34) and Fig.S25 (page S36).

Fig.R2 The typical UV/Vis spectra in different reaction systems with (a) OPD and (b) ABTS as peroxidase substrates (inset: optical image showing the corresponding color changes).

Fig.R3 (a) Schematic illustration of colorimetric detection of glucose with one-pot. (b) Normalized catalytic cascade reaction activities of Fe₁/Au NPs, Fe₁&Au NPs and Fe₁@Au NPs. (c) Absorption spectra of oxTMB with different glucose concentrations. (d) The linear calibration plots for glucose detection. (e) Selectivity evaluation of glucose detection. The concentration of glucose was 0.1 mM, and the concentrations of all interfering substances were 1 mM. (f) Recycling experiments of Fe₁@Au NPs based colorimetric glucose biosensor.

Fig.R4 (a) Schematic illustration of the smartphone-assisted sensing of glucose using all-in-one gel strips. (b) The calibration curves of the RGB value recorded by the smartphone APP toward the different concentration of glucose. (c) Application of gel-based sensor for glucose detection in real serum samples. (Note: The glucometer was used as a standard method and the values near to the circles were a coefficient variation of gel-based sensor for the same sample. All parameters were calculated from three replicates). (d) The long-term storage stability of gel-based colorimetric sensor.

Fig.R5 Specific activities of Fe₁@Au NPs and Fe₁&Au NPs.

Fig.R6 (a) Steady-state kinetic assay of GOx-like activity. (b) The corresponding Lineweaver-Burk plots with glucose as a substrate.

Comment 7: Line 164. The authors should comment what advantages present their system with respect to those reported in Refs. 21-23, and other systems recently considered in the above mentioned review (i.e., *Adv. Funct. Mater.*2021, 31, 2106023).

Response 7: We thank the reviewer for this valuable suggestion. The Refs. 21-23 in our manuscript were used to explain the obvious ESR signals of DMPO-H adducts in GOx-mimicking reaction and to verify the generation of glucose acid and the GOx-mimicking activity of Au NPs.

The conventional mixed systems involve a simple stacking of different sites by coplanar construction, such as loading Au NPs with GOx-like activity on the surface of peroxidase-mimicking nanomaterials for cascade glucose detection (*Chem. Eur. J.*,2014, 20, 7501-7506; *Adv. Mater.* 2017, 29, 1700102 and so on), but these systems cannot effectively isolate the catalytic sites and prevent interference from each other, which would accompany with a low cascade reaction efficiency.

Compared to these systems, the obtained biomimetic Fe₁@Au NPs system with unique spatial segregation sites, in which Au NPs and Fe SAs sites were separately confined and closely positioned at distinct layers of 3D nanocapsules, which can effectively facilitate H₂O₂ transfer in tandem reactions (also known as the proximity effect) and minimize H₂O₂ inhibition to POD-like activity (Fig.R7), thus can enables cascade catalytic reactions with non-interference and high efficiency. Besides, the Fe₁@Au NPs system-based biosensor shows a wide linear relationship between the absorbance of oxTMB and glucose concentrations (0-1600 μM) and low limit of detection (0.13 μM), surpassing to most of the reported cascade enzyme-mimicking catalysts-based

colorimetric glucose sensors (Table R1, some literatures considered in the above mentioned review have been included).

The above mentioned literature (Adv. Funct. Mater.2021, 31, 2106023) has been included in References list of the manuscript (references [13]).

The related description and discussion were showed in line 61-65 (page 2-3), line 225-235 (page 8) and line 239-243 (page 8-9) of the manuscript. The corresponding image (Fig.R7) and table (Table R1) were shown in supporting information as Fig.S26 (in the page S37) and Table S5 (page S46).

Fig.R7 The absorption changes of TMB at varying glucose concentrations. oxTMB generation by Fe₁/Au NPs, Fe₁&Au NPs, Fe₁@Au NPs after reacting with different glucose concentrations and 0.6 mM TMB for 1 h.

Table R1. Bifunctional oxidase-peroxidase mimicking nanozymes operating in cascade catalysis for colorimetric glucose detection.

Catalysts	Reagents/Probe	Linear range (μM)	LOD (μM)	Ref
Fe ₁ @Au NPs	TMB	0-1600	0.13	This work
Au NPs/Cu-TCPP	TMB	10-300	8.5	8
AKCN	TMB	-	0.8	15
Au/V ₂ O ₅	ABTS	0-10	0.5	19
Au ₁ /CeO ₂	TMB	10-100	10	20
Au@Pt	OPD	45-400	45	21
MnO ₂	TMB	5-1200	3.3	22
Au NPs-Ag NPs	-	5-70	3	23
Ag-Au NC-30@CeO ₂	TMB	20-240	20	24
Au@BSA-GO	TMB	10-300	0.6	25
AuPd-NE aerogels	TMB	30-250	10	26

Comment 8: Lines 165-169. These sentences express the concept of cascade catalysis, repeatedly stated by the authors. Please avoid it, and just go to the further results obtained.

Response 8: Thank you for the reviewer's kinder reminder. The repeatedly stated sentences of "As shown in Figure 3a and Figure S17, GOx-like Au NPs catalyze oxidation of glucose to generate gluconic acid and H₂O₂, and the produced H₂O₂ serves as a substrate for POD-like Fe SAs, which catalyzes the oxidation of TMB to yield the colored product oxTMB." have been deleted in our revised manuscript.

Comment 9: Line 197. "The based biosensor.". What does it mean?

Response 9: Thanks for the good comment. We apologize for indistinct description of the biosensor. "The based biosensor." means that "The Fe₁@Au NPs system based colorimetric biosensor".

The revised description was stated in line 239-240 of page 8.

Comment 10: Lines 203-206. SEM, TEM and other surface techniques cannot provide information on "bioactivity".

Response 10: Thank you for the careful reading and valuable comment. The bioactivity of the biomimetic Fe₁@Au NPs system was studied by the cycle tests. As shown in Fig. R8, the biomimetic Fe₁@Au NPs system exhibits no significant loss of bioactivity for 10 cycle tests, indicating distinguished catalytic stability. Also, no obvious changes in SEM, TEM, HAADF-STEM, and EDS mappings after 10 catalytic cycles, implying a high structural stability of Fe₁@Au NPs (Fig.R9).

The related description was shown in line 247-251 of page 9. The corresponding images (Fig.R8-R9) were shown in manuscript as Fig.4f (page 17) and supporting information as Fig.S28 (page S39).

Fig.R8 Recycling experiments of Fe₁@Au NPs based colorimetric glucose biosensor.

Fig.R9 (a) TEM, (b) SEM, (c) AC HADDF-STEM, and (d) EDS elemental mapping images of Fe₁@Au NPs after the catalytic recycling tests.

Comment 11: Line 209. It is not clear to me what do the authors mean with “To further realize the instrument-free, ... (?)”

Response 11: The “instrument-free” means that the target molecules glucose can be detected and quantified with RGB pixel values recorded using a smartphone without the need of large optical spectrum testing instruments (such as THERMO Varioskan Flash spectrophotometer, UV-vis spectrophotometer and so on). To best of our knowledge, the related description of “instrument-free” has been used in reported literatures (Anal. Chem. 2019, 91, 9292-9299 and ACS Appl. Mater. Interfaces 2020, 12, 12962–12971). To avoid misleading, we have removed the related description of “instrument-free” in the revised manuscript.

Comment 12: Lines 213-217. It is not clear how the gel filled with Fe₁@Au NPs works. Specifically, how did the authors include glucose in the gel? How does the color recognition application work? How long does it take from the addition of glucose to give the optical response? How stable is the sensor? This information should be provided not only for the gel/Fe₁@Au NPs system, but also generally for any material employed in the manuscript.

Response 12: Thanks for the reviewer’s kind comments. We are very sorry for our unclearly explanation.

(1) The working principle of the gel filled with Fe₁@Au NPs was as follows: When glucose was added onto the gel strip, Fe₁@Au NPs system catalyze the oxidation of glucose to generate gluconic acid and H₂O₂ in the presence of O₂, then the produced H₂O₂ serves as a substrate for POD-like Fe SAs catalyzes the oxidation of TMB to yield the blue product oxTMB, which leads to a colored gels.

(2) The glucose with different concentrations (0-10 mM) were added dropwise on the prepared gel strips.

(3) The working principle of the color recognition application was as follows: the color

intensity increased with increasing assay solution color, which in turn depended on the glucose concentration. Free app named ColorDesk is a portable color digitizer app that makes it easy to obtain RGB value of the gel strips in real-time. The RGB value is a mix of three color components. R is red, G is green, and B is blue. The intensity values for each RGB channel vary from 0 to 255, and the larger the value, the brighter of the color is. The zero value means the strongest intensity of that color (dark color), while the 255 value corresponds to the lowest intensity (white color). Therefore, a solution with intense color (high glucose concentration) has low RGB values, and vice versa.

(4) It takes 30 min from the addition of glucose to give the optical response.

(5) The long-term stability of gel-based colorimetric sensor was examined a time per day for 30 days. As shown in Fig.R11, 96.3% of its initial response was retained after 30 days. This indicates that the prepared gel-based colorimetric sensor possesses good stability.

(6) As suggested, the above mentioned information has been included in other system (such as solution/Fe₁@Au NPs system). The detailed description can be seen in supporting information.

Besides, the schematic illustration of the smartphone-assisted sensing of glucose using all-in-one gel strips was showed in Fig.R10. Also, we have added the detailed experiments process and data analysis about gel-based sensor in supporting information. The detailed descriptions were as follows:

Gel-based sensor of glucose detection: The glucose with different concentrations (0 - 10 mM) were added dropwise on the prepared gel strips. After 30 min incubation, the gel strips were insert into the homemade smartphone colorimetric box for RGB color picking. All the gels were recorded under the same procedure and conditions. Free app named ColorDesk is a portable color digitizer app that makes it easy to obtain RGB value of the gel strips in real-time without the need of image analysis software transformation and process. The RGB value is a mix of three color components. R is red, G is green, and B is blue. Each of the color components (R, G, and B) is represented by the range of decimal numbers from 0 to 255 (256 levels for each color).

Data analysis: The principle of detection was as follows: the color intensity increased with increasing assay solution color, which in turn depended on the glucose concentration. According to RGB color space, any color can be decomposed into red, green and blue (RGB).The intensity values for each RGB channel vary from 0 to 255, and the larger the value, the brighter of the color is. The zero value means the strongest intensity of that color (dark color), while the 255 value corresponds to the lowest intensity (white color). Therefore, a solution with intense color (high glucose concentration) has low RGB values, and vice versa. To determine the most suitable relationship for the quantification of a scanned image, we analyzed different quantitative relationships, including R, G, and B. Among all the quantitative relationships studied, G strongly correlated with the glucose concentration. Therefore, the intensity of G was selected as the analytical signal for detection.

The related discussion and description were showed in line 269-270 (page 9) of manuscript and page S9-S10 of supporting information. The corresponding images (Fig.R10-R11) was added in manuscript as Fig.5a and Fig.5d (page 18).

Fig.R10 Schematic illustration of the smartphone-assisted sensing of glucose using all-in-one gel strips.

Fig.R11 The long-term storage stability of gel-based colorimetric sensor.

Comment 13: Lines 222-245. DFT results should be presented and discussed earlier.

Response 13: We thank the reviewer for the enlightening comments. As suggested, DFT results have been presented and discussed earlier in our revised manuscript.

The DFT results was shown in line189-212 (page 7) of the revised manuscript.

Reviewer #2: The construction of artificial biomimetic systems to ensure cascade reactions with non-interfering and high efficiency, as in living cellular systems, is a great challenge. In this manuscript, Wu et al. ingeniously developed a cell-inspired design of a cascade catalysis system with highly spatially separated sites, where sites can catalyze independently and work synergistically to realize highly efficient cascade glucose detection. The novel construction method solves the interference between reactions in cascade catalysis, which would greatly drive the development of biomimetic catalysis. In practice, the performance of the biomimetic system is fairly good in both colorimetric and gel-based glucose sensors. In addition, the nanostructure of the nanocapsules was systematically characterized by state-of-the-art characterization methods, and the catalytic reaction and its mechanism were further studied by DFT calculations. In summary, this is interesting and ground-breaking work. This work is recommended to be published in Nature Communications after minor revision. However, some critical issues should be addressed.

Response: We are grateful to the Reviewer #2 for his/her recognition and appreciation of our work. We also appreciate his/her careful reading of our manuscript, enlightening comments and valuable suggestions. We have answered all questions from the Reviewer #2. The specific replies and modifications are listed as follows:

Comment 1: In this work, the spatial separation between Fe SAs and Au NPs is important for highly efficient cascade catalysis. How do the authors achieve the highly spatially separated distribution of these different sites?

Response 1: We thank the reviewer for this valuable suggestion. Fig.R12 presents a schematic of the fabrication of the biomimetic catalysis system with the highly spatially separated distribution of different sites. Fe_2O_3 was firstly synthesized by a hydrothermal method. A polydopamine (PDA) layer was subsequently deposited onto the surface to form a core-shell architecture, denoted as $\text{Fe}_2\text{O}_3@\text{PDA}$. Then, the $\text{Fe}_2\text{O}_3@\text{PDA}$ was annealed under an argon (Ar) atmosphere, during which the PDA layer was transformed in situ to form N-doped carbon shells after carbonization. Meanwhile, the Fe_2O_3 on the surface is gradually reduced to metal Fe by carbon, accompanied with the Fe atoms diffused locally in the heat drive and trapped by N defects on NC shells due to the strong interaction. Acid leaching was further introduced to remove the remained Fe_2O_3 that without thermal diffusion on the $\text{Fe}_2\text{O}_3@\text{PDA}$, and the stable Fe single atoms anchored on the inner surface of NC shell were obtained (denoted as Fe_1/NC). Finally, Au nanoparticles with an average diameter of 3.6 nm were uniform loaded on outer surface of the synthesized nanocapsule by reduction of HAuCl_4 with NaBH_4 , and the biomimetic catalysis system with the highly spatially separated distribution of different sites was obtained.

The related description was showed in line 87-107 (page 3-4) of the manuscript. The corresponding image (Fig.R12) was shown in manuscript as Fig.1a (page 14).

Fig.R12 Schematic illustration of the synthetic route of Fe₁@Au NPs.

Comment 2: Why do authors select Fe SAs as POD mimics, instead of other metal SAs (such as Cu, Co, Ni, ...)? Is there anything special advantage about Fe SAs?

Response 2: Thanks for the good comment. To verify the POD-like catalytic activity of different metal SAs, the catalytic conversion TMB to oxTMB was conducted in the presence of H₂O₂. As shown in Fig.R13a, compared with Mn, Co, Ni and Cu SAs, Fe SAs shows the highest POD-like activity. Besides, natural horseradish peroxidase (HRP) contains specific Fe metal ions that act as active sites (Fig.R13b, Anal. Chem., 2019, 91, 11994–11999). Moreover, Fe SAs with FeN₄ configuration has the similar metal center structure to natural enzyme. Thus, we selected the Fe SAs with FeN₄ configuration, instead of other metal SAs (such as Cu, Co, Ni and so on) as the active center in peroxidase-mimicking reactions.

As far as we know, the related literatures have been reported to simulate the active center of natural enzyme to enhance the enzymes-mimicking catalytic activity (Sci. Adv., 2019, 5, eaav5490 and J. Am. Chem. Soc., 2020, 142, 36, 15569–15574).

Fig.R13 (a) Comparison of the POD activity of Fe, Mn, Co, Ni and Cu SAs. (b) Active sites of HRP and Fe SAs.

Comment 3: In cascade reactions, the integration of different kinds of nanozymes in one system would accompany a low cascade reaction efficiency. Why, in this work, the biomimetic cascade catalysis system is so efficient?

Response 3: We thank the reviewer for the careful reading of our manuscript and enlightening suggestions. The efficient performance of biomimetic cascade catalysis system can be attributed to unique spatial segregation sites of Fe₁@Au NPs, in which Au NPs and Fe SAs sites were separately confined and closely positioned at distinct layers of 3D nanocapsules, which can effectively facilitate H₂O₂ transfer in tandem reactions (also known as the proximity effect) and minimize H₂O₂ inhibition to POD-like activity (Fig.R14), thus can greatly enhance the cascade catalytic activity in glucose detection.

The related description and discussion were showed in line 225-230 (page 8) of the manuscript. The corresponding image (Fig.R14) was shown in supporting information as Fig.S26 (in the page S37).

Fig.R14 The absorption changes of TMB at varying glucose concentrations. oxTMB generation by Fe₁/Au NPs, Fe₁&Au NPs, Fe₁@Au NPs after reacting with different glucose concentrations and 0.6 mM TMB for 1 h.

Comment 4: In page 4, the authors referred “Acid leaching was further introduced to remove the remained Fe₂O₃ ...”, however, “Acid leaching” had not be stated in the whole text. Besides, Fe₂O₃ was removed by acid leaching, but why Fe SAs can be remained? The relative statement about the detailed changes of the sample in this process should be supplied.

Response 4: The acid used for etching was stated in Fig.1a and page 4 of the original manuscript. The detail information about acid leaching can be seen in the “Synthesis of peanut-shaped Fe₁/NC” section (supporting information, page S4). In the acid leaching process, the metal oxides were removed by immersing the samples in the 5 M HCl solution for 6 h at 80 °C.

In pyrolysis process, the PDA layer was transformed in situ to form N-doped carbon shells. Meanwhile, the Fe₂O₃ on the surface is gradually reduced to metal Fe by carbon, accompanied with the Fe atoms diffused locally in the heat drive and trapped by N defects on NC shells to form stable Fe-N_x structure due to the strong interaction. Thus, the Fe SAs can be remained after acid leaching. This is in good agreement with the previously reported literature (J. Am. Chem. Soc. 2017, 139, 10976-10979).

The detailed changes of the sample were studied by XRD experiments. Fig.R15 shows the characteristic peaks of α -Fe₂O₃ at the initial stage. After the PDA coating and subsequent carbonization treatment, the depressed signal for α -Fe₂O₃ and the appearance of γ -Fe₂O₃ indicate the phase transition of Fe₂O₃ occurred at high temperatures. After acid etching, the typical signal of γ -Fe₂O₃ vanished and the obtained Fe₁/NC shows a wide peak at $2\theta = 25.3^\circ$ for the carbon (002) peak. Additionally, no characteristic Fe peaks can be detected, excluding the existence of large Fe metal and oxide aggregation in Fe₁/NC. Subsequently, after the deposition of Au NPs, the obtained Fe₁@Au NPs shows the typical signal of Au NPs, in which the signals at 38.2° , 44.4° , 64.5° , and 77.5° correspond to the (111), (200), (220) and (311) lattice planes (JCPDS No. 04-0784).

The related description and discussion were showed in line 87-102 (page 3-4) and line118-128 (page 4-5) of the manuscript, and the section of “Synthesis of peanut-shaped Fe₁/NC” of supporting information (page S4). The corresponding image (Fig.R15) was shown in manuscript as Fig.2a (in the page 15).

Fig. R15 XRD patterns.

Comment 5: In Figure 2a, there is a broad peak in Fe₁/NC, but the explanation is lost.

Response 5: We thank the reviewer for this valuable suggestion. The Fe₁/NC with a wide peak at $2\theta = 25.3^\circ$ can be assigned to the carbon (002) peak. The explanation of the broad peak was added in Fig.R15.

The related explanation was shown in line 123-124 (page 4-5) of manuscript. The corresponding image (Fig.R15) was shown in the revised manuscript as Fig.2a (in the page 15).

Comment 6: The EPR spectra presented in Figure S15 probably have different line widths, which may call into question assertions about the relative catalytic activity, unless the intensities were obtained by careful integration and baseline corrections.

Response 6: The EPR experiment was performed in Experimental Center of Physics and Chemistry, University of Science and Technology of China, which was checked by the instrument engineer (Jiafu Chen). The shown EPR spectra were obtained by careful integration and baseline corrections.

Comment 7: I noticed that the authors use the APP to realize real-time and visual determination of glucose displayed in Figure 3g, and relate RGB values and concentration of glucose, what's the process? Also, how to get the RGB values by the APP?

Response 7: Thanks for the good comment. The specific process of gel-based sensor for glucose detection was shown as bellow. Free app named ColorDesk is a portable color digitizer app that makes it easy to obtain RGB value of the gel strips without the need of image analysis software transformation and process.

Gel-based sensor of glucose detection: The glucose with different concentrations (0 - 10 mM) were added dropwise on the prepared gel strips. After 30 min incubation, the gel strips were insert into the homemade smartphone colorimetric box for RGB color picking. All the gels were recorded under the same procedure and conditions. Free Android app ColorDesk is a portable color digitizer app that makes it easy to obtain RGB value of the gel strips in real-time without the need of image analysis software transformation and process. The RGB value is a mix of three color components. R is red, G is green, and B is blue. Each of the color components (R, G, and B) is represented by the range of decimal numbers from 0 to 255 (256 levels for each color).

Data analysis: The principle of detection was as follows: the color intensity increased with increasing assay solution color, which in turn depended on the glucose concentration. According to RGB color space, any color can be decomposed into red, green and blue (RGB).The intensity values for each RGB channel vary from 0 to 255, and the larger the value, the brighter of the color is. The zero value means the strongest intensity of that color (dark color), while the 255 value corresponds to the lowest intensity (white color). Therefore, a solution with intense color (high glucose concentration) has low RGB values, and vice versa. To determine the most suitable relationship for the quantification of a scanned image, we analyzed different quantitative relationships, including R, G, and B. Among all the quantitative relationships studied, G strongly correlated with the glucose concentration. Therefore, the intensity of G was selected as the analytical signal for detection.

The detailed experimental process and data assay can be seen in supporting information as the section of “Gel-based sensor of glucose detection” and “Data analysis” (in the page S9-S10).

Reviewer #3: This manuscript reports Cell-inspired nanozyme for biomimetic cascade catalysis toward glucose and its application for glucose detection. The results and experimental data appear to be satisfactory. However, there are certain issues that must be resolved to improve the quality of the manuscript. The followings are the suggestions for improving its quality.

Response: We are grateful to the Reviewer #3 for his/her recognition and appreciation of our work. We also appreciate his/her careful reading of our manuscript, enlightening comments and valuable suggestions. We have answered all questions from the Reviewer #3. The specific replies and modifications are listed as follows:

Comment 1: There is no experimental details for selectivity test in Fig S22. How much concentration of glucose and interfering substances? What experimental method was used to get and evaluate the activity toward glucose and substances? In addition, galactose and lactose (highly possible interfering substances) should be investigated for selectivity test. Note that the applied concentration of interfering substances for selectivity should be based on their physiological level.

Response 1: Thanks for the reviewer's kind comments. We are very sorry for missing the experimental details for selectivity test in Fig S22, which may cause confusion for readers to understand our work.

(1) The experimental details for selectivity test was as follows. The selectivity of Fe₁@Au NPs system for glucose was detected in solution containing glucose (**0.1 mM**) or interfering substances (dopamine (DA), L-cysteine, sucrose, fructose, ascorbic acid (AA), uric acid (UA), maltose, lactose and galactose, **1 mM**). Specifically, the Fe₁@Au NPs (20 μ L, 200 μ g/ml) and glucose (20 μ L, 1 mM) or other interfering substances (20 μ L, 10 mM) were added into 96-well plates containing PBS buffer (140 μ L, 10 mM, pH 7.2). Then the mixed solution was incubated for 30 minutes at 37 °C. Next, the TMB (20 μ L, 6 mM) was introduced to the above solution. Finally, the mixture was incubated for 10 min and detected at 652 nm using a THERMO Varioskan Flash spectrophotometer.

(2) The colorimetry (CM) was used to get and evaluate the activity toward glucose and substances.

(3) As suggested, we added the selectivity test of interfering substances (galactose and lactose), as shown in Fig.R16. It's worth noting that the concentrations of all interfering substances were 1 mM in selectivity tests, which were higher than their physiological level.

The description of experimental details for selectivity test was added in supporting information as the section of "The selectivity evaluation of glucose detection" (page S8). The corresponding image (Fig.R16) was added in manuscript as Fig.4e (page 17).

Fig. R16 Selectivity evaluation of glucose detection.

Comment 2: Give the experimental details for Fig S23.

Response 2: Thanks for the good comment. We are very sorry for missing the experimental details for Fig S23. The experimental details for Fig S23 was as follow.

Recycling experiments: In the recycling experiments, the Fe₁@Au NPs (20 μL, 200 μg/ml) and glucose (20 μL, 10 mM) were added into a tube containing PBS buffer (140 μL, 10 mM, pH 7.2). Then the mixed solution was incubated for 30 minutes at 37 °C. Then, the TMB (20 μL, 6 mM) was introduced to the above solution. The mixture was incubated for 10 min and detected at 652 nm using a THERMO Varioskan Flash spectrophotometer. Next, the supernatant was removed after the test, and a new portion of glucose and PBS buffer and TMB were added. The mixture was incubated and detected at 652 nm using a THERMO Varioskan Flash spectrophotometer. Recycling continued for ten runs. All reactions were duplicated.

Storage stability tests: The long-term stability of Fe₁@Au NPs based glucose biosensor was evaluated by measuring the response towards the same glucose concentrations for every few days. Specifically, the Fe₁@Au NPs (20 μL, 200 μg/ml) and glucose (20 μL, 10 mM) were added into 96-well plates containing PBS buffer (140 μL, 10 mM, pH 7.2). Then the mixed solution was incubated for 30 minutes at 37 °C. Next, the TMB (20 μL, 6 mM) was introduced to the above solution. Finally, the mixture was incubated for 10 min and detected at 652 nm using a THERMO Varioskan Flash spectrophotometer.

The related experimental details have been added in supporting information as the section of “Recycling experiments” and “Storage stability tests” (in the page S8-S9). The corresponding images (Figure S23) were transferred to the revised manuscript as Fig.4f (page 17) and supporting information as Fig.S27 (page S38).

Comment 3: The authors described that “When glucose was added onto the gel strip, the gel colors changed from colorless to blue due to TMB oxidation by the using the generated H₂O₂ as a mediate substrate.”. The resulting blue color was demonstrated in Fig 3(f) and (h). However, contradictively, authors chose Green value of RGB for quantitative analysis instead of Blue value. Experiments and data analysis involved in

colorimetric assay are unclear and vague.

Response 3: We apologize for our unclear explanation of experiments and data analysis, which cause difficulties for readers to understand our work. The principle of detection was as follows: the color intensity increased with increasing assay solution color, which in turn depended on the glucose concentration. According to RGB color space, any color can be decomposed into red, green and blue (RGB). The intensity values for each RGB channel vary from 0 to 255, and the larger the value, the brighter of the color is. The zero value means the strongest intensity of that color (dark color), while the 255 value corresponds to the lowest intensity (white color). Therefore, a solution with intense color (high glucose concentration) has low RGB values, and vice versa.

To determine the most suitable relationship for the quantification of a scanned image, we analyzed different quantitative relationships, including R, G, and B. As shown in Fig.R17, a good linear relationship was established between the RGB values and glucose concentrations in the range of 1–10 mM ($R^2 = 0.990$), 0.1–10 mM ($R^2 = 0.998$), and 0.4–10 mM ($R^2 = 0.993$) when R, G, and B were employed as the detection signal, respectively. Among all the quantitative relationships studied, G strongly correlated with the glucose concentration. Therefore, the intensity of G was selected as the analytical signal for detection. The detailed experiments process and data analysis were as follows.

Gel-based sensor of glucose detection: The glucose with different concentrations (0 - 10 mM) were added dropwise on the prepared gel strips. After 30 min incubation, the gel strips were insert into the homemade smartphone colorimetric box for RGB color picking. All the gels were recorded under the same procedure and conditions. Free app named ColorDesk is a portable color digitizer app that makes it easy to obtain RGB value of the gel strips in real-time without the need of image analysis software transformation and process. The RGB value is a mix of three color components. R is red, G is green, and B is blue. Each of the color components (R, G, and B) is represented by the range of decimal numbers from 0 to 255 (256 levels for each color).

Data analysis: The principle of detection was as follows: the color intensity increased with increasing assay solution color, which in turn depended on the glucose concentration. According to RGB color space, any color can be decomposed into red, green and blue (RGB). The intensity values for each RGB channel vary from 0 to 255, and the larger the value, the brighter of the color is. The zero value means the strongest intensity of that color (dark color), while the 255 value corresponds to the lowest intensity (white color). Therefore, a solution with intense color (high glucose concentration) has low RGB values, and vice versa. To determine the most suitable relationship for the quantification of a scanned image, we analyzed different quantitative relationships, including R, G, and B. Among all the quantitative relationships studied, G strongly correlated with the glucose concentration. Therefore, the intensity of G was selected as the analytical signal for detection.

The detailed experiments process and data analysis have been added in supporting information as the section of “Gel-based sensor of glucose detection” and “Data

analysis” (in the page S9-S10). The related discussion and description were added in page S40 of the supporting information. The corresponding image (Fig.R17) was showed in manuscript as Fig.5b (in the page 18) and Fig.S29 (page S40).

Fig.R17 Plots of the values of (a) R, (b) G, and (c) B versus the glucose concentration, respectively.

Comment 4: What kind of color recognizer application (APP) did you use? Did you develop it? The authors should provide experimental details unfeignedly.

Response 4: Thanks for the good comment. The color recognizer application (APP) named ColorDesk is a free app. The experimental details can be seen the Response 3. As suggested, the related experimental details have been added supporting information as the section of “Gel-based sensor of glucose detection” and “Data analysis” (in the page S9-S10).

Comment 5: The interference from colored erythrocytes (red blood cells) present in whole blood would seriously affect the interpretation results. The authors have the solution to deal with this problem? The authors need to add the discussion on it.

Response 5: We thank the reviewer for the careful reading of our manuscript and enlightening suggestions. We agree with the reviewer that the colored erythrocytes present in whole blood would interfere in the chromogenic reaction. Thus, separation of serum or plasma from whole blood by a centrifuge or a magnetic method is generally required prior to testing in diagnostic labs.

In our study, the obtained agarose-based gels (1% w/v) possess a porous structure with the size of nanometer scale (Fig.R18), which can allow small molecules (glucose) to pass through freely while preventing the entry of biological macromolecules (red blood cells, 7-8 μm diameter). When the whole blood sample was loaded into the prepared gel, small target molecules (glucose) pass through and were detected via a chromogenic reaction, while the red blood cells remained on the front of the gel. A smartphone was used to record the color signal at the side of the gel to achieve quantitative determination of the targets without any interference from the blood samples (Fig.R19). The related literature has also been reported (Anal. Chem. 2021, 93, 14755-14763).

The related discussion was added in page S41 of the supporting information. The corresponding images (Fig.R18 and Fig.R19) were added in Fig.S30 (page S41).

Fig.R18 The pore-size distribution of agarose-based gel.

Fig.R19 The detection of glucose in whole blood.

Comment 6: The authors should give the real glucose test (e.g. human serum or blood) and the comparison with glucometer to demonstrate the feasibility of gel-based colorimetric sensing system.

Response 6: Thanks for the good comment. As suggested, some real serum samples contributed by volunteers were used to further evaluate the application of the gel-based sensors in disease diagnosis. As shown in Fig.R20, the measurement results of the gel-based sensor exhibits a high correlation with the levels measured by the commercial glucometer, which could also be verified by favorable coefficients of variation (0.036-0.064), indicating that the gel-based sensor was able to serve as potential analytical platforms in disease diagnosis.

The related discussion and description were added in line 264-269 (page 9) of the manuscript and the section of “Serum samples analysis” (page S10) of the supporting information. The corresponding image (Fig.R20) was added in manuscript as Fig.5c (page 18).

Fig.R20 Application of gel-based sensor for glucose detection in real serum samples. (Note: The glucometer was used as a standard method and the values near to the circles were a coefficient variation of gel-based sensor for the same sample. All parameters were calculated from three replicates).

Comment 7: Aside from the stability of Fe₁@Au NPs shown in Fig S23, the long-term stability of gel-based colorimetric sensor should be investigated for practical applications.

Response 7: We thank the reviewer for the careful reading of our manuscript and enlightening suggestions. The long-term stability of gel-based colorimetric sensor was examined a time per day for 30 days. As shown in Fig.R21, 96.3% of its initial response was retained after 30 days. This indicates that the prepared gel-based colorimetric sensor possesses good long-term stability.

The related discussion and description were showed in line 269-270 (page 9) of the manuscript. The corresponding image (Fig.R21) was added in manuscript as Fig.5d (in the page 18).

Fig.R21 The long-term storage stability of gel-based colorimetric sensor.

REVIEWERS' COMMENTS

Reviewer #1 (Remarks to the Author):

The authors have addressed most of my comments and, in my opinion, the manuscript is now suitable for publication, at least from a scientific point of view. However, there are still some minor problems with the English. Some examples are shown below.

Lines 73-74. The term "which", after the comma, is not appropriate. The subject is "Fe SAs" and not "thermal diffusion".

Line 157. The use of "Since" is misleading. The sentence is better read with the incipit : "As the catalysis system (Fe₁@Au NPs) includes... "

Lines 162 - 165. Rephrase the sentences. Put a full stop after O₂. Delete "in which" and start the sentence with "The generated..." or "The latter species...".

Line 176. "...we also prepared a conventional mixing system...:" and "... a coplanar construction system..." One can use a convention mixing system for the preparation..., or can prepare a coplanar system...

And so on..

Reviewer #2 (Remarks to the Author):

The authors have revised the manuscript according to reviewers' comments and it is recommended to be accepted at the current stage.

Reviewer #3 (Remarks to the Author):

The manuscript has been sincerely revised based on raised comments. Almost issues have been addressed. The current version can be accepted after minor revision.

- In order to make experiments clear,

1) give the applied volume of sample on gel-based sensor.

In supporting: "The glucose with different concentrations (0-10 mM) were added dropwise on the prepared gel strips. "

2) the resulting color intensity is not affected as applied volume increase?

3) give the storage condition of gel-based sensor for long-term storage stability (Fig. R21).

- There is no information on IRB NO.

In supporting: "All procedures are approved by Institutional Ethics Review Committee."

Response to referees and revisions made in the manuscript:

We thank the reviewers for their helpful and constructive comments and suggestions. We have responded to each and every question as detailed below.

Reviewer #1: The authors have addressed most of my comments and, in my opinion, the manuscript is now suitable for publication, at least from a scientific point of view. However, there are still some minor problems with the English. Some examples are shown below.

Response: We are grateful to the Reviewer #1 for his/her recognition and appreciation of our work. We also appreciate his/her careful reading of our manuscript, enlightening comments and valuable suggestions. We have answered all questions from the Reviewer #1. The specific replies and modifications are listed as follows:

Comment 1: Lines 73-74. The term “which”, after the comma, is not appropriate. The subject is “Fe SAs” and not “thermal diffusion”.

Response 1: We thank the reviewer for the careful reading of our manuscript and enlightening suggestions. The sentence of “The monodispersed Fe SAs are fixed on the interior surface of the nanocapsules through thermal diffusion, which shows peroxidase (POD)-like activity.” was corrected as “The monodispersed Fe SAs fixed on the interior surface of the nanocapsules through thermal diffusion show peroxidase (POD)-like activity.”

Comment 2: Line 157. The use of “Since” is misleading. The sentence is better read with the incipit : “As the catalysis system (Fe₁@Au NPs) includes... “

Response 2: Thanks for the good comment. In order to further improve the quality and readability of the manuscript, “Since constructed the catalysis system (Fe₁@Au NPs) includes different catalytic metal species” was revised to “As the catalysis system (Fe₁@Au NPs) includes different catalytic metal species”.

Comment 3: Lines 162 - 165. Rephrase the sentences. Put a full stop after O₂. Delete "in which" and start the sentence with "The generated..." or “The latter species...”.

Response 3: We thank the reviewer for this valuable suggestion. As suggested, “Firstly, the Fe₁@Au NPs system can catalyze the oxidation of glucose to generate gluconic acid and H₂O₂ in the presence of O₂, in which the generated gluconic acid and H₂O₂ were verified by a specific colorimetric assay” was revised to “Firstly, the Fe₁@Au NPs system can catalyze the oxidation of glucose to generate gluconic acid and H₂O₂ in the presence of O₂. The generated gluconic acid and H₂O₂ were verified by a specific colorimetric assay”.

Comment 4: Line 176. “...we also prepared a conventional mixing system...:” and “... a coplanar construction system...” One can use a convention mixing system for the preparation..., or can prepare a coplanar system...

And so on..

Response 4: Thanks for the good comment. As suggested, “For a better comparison,

we also prepared a conventional mixing system by physically mixing Fe SAs with Au NPs sites (denoted as Fe₁/Au NPs) and a coplanar construction system according to the previous reports (denoted as Fe₁&Au NPs)” was revised to “For a better comparison, one can use a convention mixing system for the preparation by physically mixing Fe SAs with Au NPs sites (denoted as Fe₁/Au NPs), or can prepare a coplanar system according to the previous reports (denoted as Fe₁&Au NPs)”.

Furthermore, we have carefully checked every sentence and corrected grammatical issues in manuscript.

1. “kind” in page 2 was corrected as “type”.
2. The sentence of “For example, glucose oxidase (GOx)-like and horseradish peroxidase (HRP)-like active sites were combined to obtain hybrid cascade catalysts that have been employed in glucose detection” in page 2 was corrected as “For instance, hybrid cascade catalysts that combine glucose oxidase (GOx)-like and horseradish peroxidase (HRP)-like active sites have been developed and utilized in glucose detection.”.
3. “Inspired by eukaryotic cells, we developed a cell-stimulated design of a biomimetic cascade catalysis system as showed in Fig. 1a, in which Fe SAs and Au NPs are separately confined and closely positioned at distinct layers of 3D nanocapsules to realize efficient cascade catalysis.” in page 3 was corrected as “Inspired by eukaryotic cells, we developed a cell-stimulated design of a biomimetic cascade catalysis system as illustrated in Fig. 1a. In this system, Fe SAs and Au NPs are separately confined and closely positioned at distinct layers of 3D nanocapsules to realize efficient cascade catalysis.”.
4. “at” in page 4 was corrected as “in”.
5. “for” in page 4 was corrected as “corresponding to”.

In addition, the whole manuscript was polished by Nature Author Services again to ensure linguistic issues resolved. The revised parts are highlighted in blue in both the revised manuscript and supplementary information.

Reviewer #2: The authors have revised the manuscript according to reviewers' comments and it is recommended to be accepted at the current stage.

Response: We thank the Reviewer #2 very much for the positive comments.

Reviewer #3: The manuscript has been sincerely revised based on raised comments. Almost issues have been addressed. The current version can be accepted after minor revision.

Response: We are grateful to the Reviewer #3 for his/her recognition and appreciation of our work. We also appreciate his/her careful reading of our manuscript, enlightening comments and valuable suggestions. We have answered all questions from the Reviewer #3. The specific replies and modifications are listed as follows:

Comment: - In order to make experiments clear,

1) give the applied volume of sample on gel-based sensor.

In supporting: "The glucose with different concentrations (0-10 mM) were added dropwise on the prepared gel strips."

2) the resulting color intensity is not affected as applied volume increase?

3) give the storage condition of gel-based sensor for long-term storage stability (Fig. R21).

- There is no information on IRB NO.

In supporting: "All procedures are approved by Institutional Ethics Review Committee."

Response: We thank the reviewer for this valuable comments. The specific replies are listed as follows:

1) The applied volume of sample on gel-based sensor was 20 μ L. The related sentence of "The glucose with different concentrations (0-10 mM) were added dropwise on the prepared gel strips." was revised to "The glucose with different concentrations (20 μ L, 0-10 mM) were added dropwise on the prepared gel strips."

2) As shown in Fig.R1, the G value decreases with the applied volume in a same glucose concentration. Thus, applied volume of sample has an effect on the RGB value, also the color intensity.

3) The storage condition of gel-based sensor was at 4°C temperature in dark.

Fig.R1 The RGB value (G) recorded by the smartphone APP toward the different volume (20 μ l, 40 μ l and 60 μ l) at a same glucose concentration (1 mM).

The related descriptions were added in the section of "Methods" (page 16 and 17) of the revised manuscript.

Besides, all procedures are approved by Institutional Ethics Review Committee of the First Affiliated Hospital of USTC (2021 KY 089). The related IRB NO was added in

section of “Human subjects and serum samples analysis” (page 18) of the revised manuscript.